# MAESTRO: LEARNING TO COLLABORATE VIA CONDITIONAL LISTWISE POLICY OPTIMIZATION FOR MULTI-AGENT LLMS

## ABSTRACT

Multi-agent systems (MAS) built on Large Language Models (LLMs) are being used to approach complex problems and can surpass single model inference. However, their success hinges on navigating a fundamental cognitive tension: the need to balance broad, **divergent** exploration of the solution space with a principled, **convergent** synthesis to the optimal solution. Existing paradigms often struggle to manage this duality, leading to premature consensus, error propagation, and a critical credit assignment problem that fails to distinguish between genuine reasoning and superficially plausible arguments. To resolve this core challenge, we propose the **Multi-Agent Exploration–Synthesis framework Through Role Orchestration (MAESTRO)**, a principled paradigm for collaboration that structurally decouples these cognitive modes. MAESTRO uses a collective of parallel Execution Agents for diverse exploration and a specialized Central Agent for convergent, evaluative synthesis. To operationalize this critical synthesis phase, we introduce **Conditional Listwise Policy Optimization (CLPO)**, a reinforcement learning objective that disentangles signals for strategic decisions and tactical rationales. By combining decision-focused policy gradients with a list-wise ranking loss over justifications, CLPO achieves clean credit assignment and stronger comparative supervision. Experiments on mathematical reasoning and general problem-solving benchmarks demonstrate that MAESTRO, coupled with CLPO, consistently outperforms existing state-of-the-art multi-agent approaches, delivering absolute accuracy gains of 6% on average and up to 10% at best.

## 1 INTRODUCTION

The rise of large language models (LLMs) have enabled a new type of *multi-agent system* (MAS) (Park et al., 2023; Chen et al., 2023a; Zhu et al., 2025), where multiple model instances collaborate to tackle problems that exceed the capacity of any single model (Zhang et al., 2024a; Qiao et al., 2024; Han et al., 2025). By distributing roles and enabling structured interaction, MASs hold the promise of achieving robustness, creativity, and reliability that emerge from collective intelligence (Cheng et al., 2024; Pezeshkpour et al., 2024). At the heart of any effective collaborative system lies a fundamental cognitive tension. Early work in the psychology of creativity (Runco & Chand, 1995; Brophy, 2001; Zhang et al., 2020) emphasizes that intelligent problem-solving requires a dynamic balance between two seemingly contradictory modes of thought: **Divergent Creativity** and **Convergent Critique**. Guilford's theory of divergent and convergent thinking (Guilford, 1967) formalizes this duality: divergence is the generative process of exploring a wide array of alternative hypotheses, while convergence is the evaluative process of comparing, refining, and synthesizing these options. Without the former, a system risks premature closure; without the latter, it risks incoherence and indecision (Sternberg & Lubart, 1991; Cropley, 2006). Achieving a principled and effective synergy between these two capabilities is the essential challenge for effective LLM agent collaboration.

Despite their diversity, the limitations of existing paradigms point to a set of recurring requirements for advancing multi-agent collaboration. First, an effective system should strike a balance between **divergent exploration and convergent synthesis**, ensuring that creativity is not stifled by premature agreement yet also not lost in unbounded search. Second, it should enable **disentangled credit assignment** across structured outputs (Li et al., 2025; He et al., 2025), so that strategic decisions and

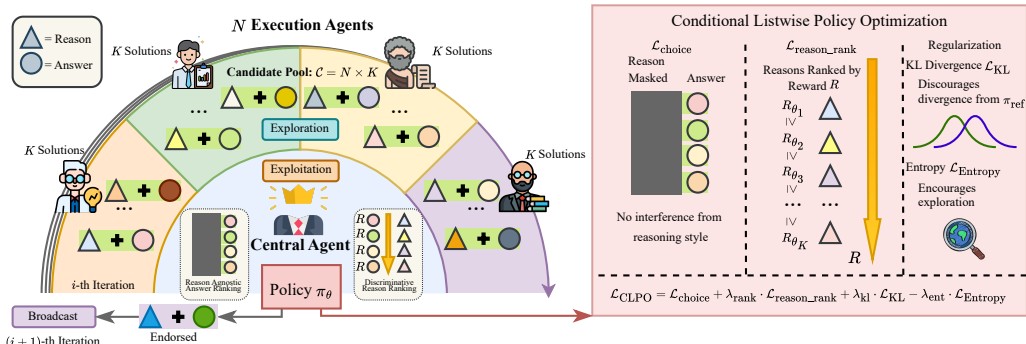

Figure 1: Overview of the **MAESTRO** framework. First, $N$ execution agents each generate $K$ candidate reasoning-answer pairs, forming a broad solution pool. A central agent then governs exploitation by applying discriminative selection over the candidate set. The decision policy $\pi_\theta$ is trained under **Conditional Listwise Policy Optimization (CLPO)**, which integrates a choice-aware objective, a reasoning-rank objective, and regularization terms including KL divergence and entropy. The endorsed candidate is subsequently broadcast for iterative refinement, enabling multi-round improvement within a principled multi-agent collaboration paradigm.

supporting rationales receive distinct and targeted learning signals rather than being conflated into a single monolithic reward. Third, a robust framework requires **transparent and scalable interaction protocols** (Qian et al., 2024; Hu et al., 2024c), where information is propagated in analyzable ways that remain efficient as the number of agents and rounds increases (Yang et al., 2025). Together, these desiderata highlight the limitations of existing approaches and motivate the need for a new paradigm that integrates principled exploration, evaluative precision, and collaborative scalability.

To address these desiderata, we propose the **Multi-Agent Exploration–Synthesis framework Through Role Orchestration (MAESTRO)**, a principled paradigm for multi-agent collaboration (Figure 1). The effectiveness of MAESTRO arises not from any single component, but from the synergistic orchestration of specialized roles. MAESTRO explicitly operationalizes the divergent–convergent duality through a structured role orchestration: (*i*) **Divergence as Collective Exploration**, where multiple Execution Agents generate a broad and diverse candidate pool; (*ii*) **Convergence as List-wise Bayesian Synthesis**, where a Central Agent evaluates these candidates to identify and endorse the most promising solution; and (*iii*) **Broadcast as Public Conditioning**, where the endorsed solution is propagated back to all agents, guiding the next round of exploration. This cycle of divergence, convergence, and broadcast structures collaboration into analyzable and scalable phases. To further optimize the convergence phase, we introduce **Conditional Listwise Policy Optimization (CLPO)**, a reinforcement learning objective that disentangles decision-making from rationale generation. Unlike standard GRPO-style sequence-level training, CLPO allocates learning signal separately to *decisions* (which candidate to endorse) and *reasons* (why this choice is defensible). MAESTRO and CLPO constitute a new paradigm for multi-agent collaboration that integrates cognitive inspiration with principled optimization. The main contributions of this paper are:

- We introduce the Multi-Agent Exploration–Synthesis framework Through Role Orchestration (MAESTRO), a principled paradigm for multi-agent collaboration that explicitly operationalizes the divergent–convergent duality through three coordinated phases.
- We propose Conditional Listwise Policy Optimization (CLPO), an RL objective that decouples signals for *decisions* and *reasons*. CLPO combines group-relative decision optimization with listwise rationale ranking for clean credit assignment and stable convergence.
- Extensive experiments on mathematical and general reasoning benchmarks show that MAESTRO with CLPO achieves significant improvements over state-of-the-art baselines.

## 2 RELATED WORK

We highlight representative related works in multi-agent LLM collaboration and RL for multi-agent LLMs. For a more in-depth account of related work, see Appendix A.

**Multi-Agent LLM Collaboration.** Large language model (LLM) based multi-agent systems have been proposed to overcome the inherent limits of single models in context length, sequential reasoning, and skill breadth (Abdelnabi et al., 2023; Wu et al., 2024; Yan et al., 2025; Dai et al., 2025). By coordinating multiple agents, these systems can decompose tasks, critique candidate solutions, and integrate diverse perspectives (Hong et al., 2023; Chen et al., 2023b; Qiao et al., 2024; Pan et al., 2024). One common design follows *prestructured coordination*, where communication topologies and protocols are fixed in advance (Chen et al., 2024; Mukobi et al., 2023; Wang et al., 2023; Abdelnabi et al., 2024). Debate and peer-review frameworks (Du et al., 2023; Chan et al., 2023; Liu et al., 2024) encourage agents to cross-examine one another, while chain or graph structures regulate message flow (Qian et al., 2024; Liu et al., 2023b). These methods reduce hallucinations and improve consistency but often enforce early convergence, limiting exploration and leaving credit assignment opaque (Hu et al., 2024a; Yue et al., 2025). A second line explores *adaptive coordination*, where the collaboration graph is reorganized dynamically during inference. Examples include routing and pruning strategies (Yue et al., 2025; Hu et al., 2024b), as well as workflow and graph-search approaches that optimize interaction structures through reinforcement or evolutionary methods (Zhuge et al., 2024; Zhang et al., 2024c;b).These frameworks improve scalability and efficiency but typically treat feedback as a global property of the entire system, which limits their ability to provide fine-grained credit assignment for individual contributions.

**Reinforcement Learning for Multi-Agent LLMs.** Reinforcement learning (RL) provides a natural mechanism for improving collaboration in multi-agent LLM systems beyond static prompt design (Madaan et al., 2023; Zelikman et al., 2024; 2022; Zhuang et al., 2024; Zhu et al., 2025). Rather than relying solely on prestructured debate or workflow rules, RL enables agents to adapt interaction patterns from feedback, learning when and how to communicate to achieve stronger group performance (Zhou et al., 2025; Wang et al., 2024; Xu et al., 2025; Wan et al., 2025; Park et al., 2025; Yang & Thomason, 2025). These approaches show that reward-driven updates can uncover strategies for dynamic role assignment, coordination, and decision aggregation. A central challenge in this setting is credit assignment (Liu et al., 2023a; Zhang et al., 2024d;e; Li et al., 2024b). Most existing methods propagate reward at the system level, treating outcomes as global properties of the entire team (Jiang et al., 2025; Lin et al., 2025). This global reward fails to identify the specific contributions of individual agents or to separate the quality of rationales from the correctness of final decisions. Recent efforts attempt to design more targeted objectives (Wei et al., 2025; Alsadat & Xu, 2024), but principled, fine-grained supervision remains limited. Our work focuses specifically on the convergence step: we view it as a structured optimization problem and design an objective that provides more precise credit assignment than existing system-level rewards.

## 3 METHODOLOGY

We introduce a novel learning paradigm to enhance the collective problem-solving capabilities of multi-agent systems. We design a collaborative process through the lens of a new structural framework, the **Multi-Agent Exploration–Synthesis (MAESTRO)** paradigm, which orchestrates the generation of diverse solutions and the subsequent critical evaluation. At the core of this framework lies our primary algorithmic contribution, **Conditional Listwise Policy Optimization (CLPO)**, a reinforcement learning algorithm specifically designed to train the central decision-making policy. This methodology systematically addresses the challenges of credit assignment and signal poverty inherent in complex, language-based collaborative tasks.

### 3.1 PRELIMINARIES

We study a round-based collaborative protocol for answering question $q$. In round $t$, each of the $N$ execution agents independently samples $K$ candidates conditioned on the current context $s_t^{(i)} := (q, b_{t-1}, z_{t-1}^{(i)})$ for $i \in [N]$, where $b_{t-1}$ denotes the previous public broadcast, and $z_{t-1}^{(i)}$ denotes the $i$-th agent's private history (state). This yields a total slate $\mathcal{C}_t$ of $N \times K$ candidate responses. The central policy then performs convergence by selecting one candidate in $\mathcal{C}_t$ to endorse and issues a public broadcast $b_t$ that contains the index of the answer and optionally a brief justification. The broadcast $b_t$ then conditions the next round $t + 1$. This process stops after a fixed number of rounds $R$, or when a stopping rule is met. Supervision is primarily the correctness of the endorsed answer at termination, and an optional term rewards the comparative quality of the justification.

## 3.2 THE MAESTRO FRAMEWORK: A PARADIGM FOR COLLECTIVE SYNTHESIS

We now formally introduce the Multi-Agent Exploration–Synthesis (MAESTRO) framework. MAE-STRO operationalizes the divergent–convergent model of creative problem-solving in a principled manner, by decomposing each round of the collaborative process into two distinct phases, which we view through the lenses of Bayesian inference and information theory.

**Phase 1: Divergence as Collective Exploration.** The primary objective of the divergence phase is to effectively explore the vast solution space, mirroring the divergent thinking process. This is achieved through a collective of $N$ parallel Execution Agents. Conditioned on the current state $s_t^{(i)}$, each agent is tasked with generating a diverse set of $K$ candidate solutions. Note here that each candidate solution is a complete trajectory, e.g., a full reasoning chain leading to a final answer. Formally, each agent $i \in [N]$ at time $t$ samples from its policy $\pi_{\phi^{(i)}}$ as follows:

$$c_{t,k}^{(i)} \sim \pi_{\phi^{(i)}}(\cdot \mid q, z_{t-1}^{(i)}, b_{t-1}), \quad k \in [K]. \tag{1}$$

This collective effort produces candidate pool $\mathcal{C}_t = \{\{c_{t,k}^{(i)}\}_{k=1}^K\}_{i=1}^N$. A key metric for this phase is the *coverage probability ($p_t$)* that the pool contains at least one correct solution:

$$p_t := \Pr\Big(\bigcup_{i=1}^N \bigcup_{k=1}^K E(c_{t,k}^{(i)}) \,\Big|\, s_t^{(1)}, \ldots, s_t^{(N)}\Big), \tag{2}$$

where $E(c)$ is the event that candidate $c$ is correct. The primary goal of Phase 1 is to increase the expected coverage $p_t$ in a fixed resource budget.

*Epsilon-greedy exploration.* To prevent over-conditioning during candidate generation, we allocate a small broadcast-agnostic exploration mass using a simple *epsilon-greedy* strategy. Specifically, we sample $\tilde{\pi}_{\phi^{(i)}}(\cdot \mid q, z_{t-1}^{(i)}, b_t) = (1-\varepsilon)\,\pi_{\phi^{(i)}}(\cdot \mid q, z_{t-1}^{(i)}, b_{t-1}) + \varepsilon\,\pi_{\phi^{(i)}}^{\text{base}}(\cdot \mid q)$ with default $\varepsilon = 0.1$, where $\pi_{\phi^{(i)}}$ is defined in (1). This yields a coverage floor: for any subset $A$ of the candidate space, $\tilde{\pi}_{\phi^{(i)}}(A \mid s_t^{(i)}) \geq \varepsilon\,\pi_{\phi^{(i)}}^{\text{base}}(A \mid q)$, so regions reachable by the base policy retain non-zero sampling mass. In practice, we implement the mixture via per-sample random dropout, using the base prompt with probability $\varepsilon$ and otherwise conditioning on the broadcast and the agent's private history.

**Phase 2: Convergence as List-wise Bayesian Synthesis.** Following divergent exploration in Phase 1, the convergence phase is orchestrated by a single Central Agent. Its role is to evaluate and synthesize the collective information in the slate $\mathcal{C}_t$. We view this step as approximating a Bayesian decision over the posterior probabilities for round $t$:

$$\eta_{t,k}^{(i)} := \Pr\big(E(c_{t,k}^{(i)}) \mid q, \mathcal{C}_t\big), \quad i \in [N],\ k \in [K]. \tag{3}$$

Recall that under a 0–1 loss, the Bayes optimal action selects $(i^\star, k^\star) \in \arg\max_{i,k} \eta_{t,k}^{(i)}$. We therefore train our Central Agent's policy, $\pi_\theta(\cdot \mid q, \mathcal{C}_t)$, to approximate this optimal Bayes decision rule via the CLPO loss (Section 3.3). The success of this phase is measured by the *identification probability ($q_t$)*, defined as the conditional probability that the policy selects a correct candidate given that the slate $\mathcal{C}_t$ contains at least one correct option. Specifically, let $S_t := \{(i,k) \in [N] \times [K] \mid E(c_{t,k}^{(i)}) \text{ holds}\}$ be the latent set of correct candidates and let $(i_t, k_t) \sim \pi_\theta(\cdot \mid q, \mathcal{C}_t)$ denote the centralized decision. Then the identification probability $q_t$ is defined as:

$$q_t := \Pr\big((i_t, k_t) \in S_t \mid q, \mathcal{C}_t, \{|S_t| \geq 1\}\big). \tag{4}$$

This metric quantifies the agent's critical evaluation and synthesis capability.

**Broadcast as Public Conditioning.** After selection in Phase 2, the Central Agent emits a public broadcast $b_t$, containing the endorsed index and a compact justification. This broadcast $b_t$ conditions the next round $t + 1$. We can interpret the flow of information as reducing the Shannon entropy of the ground-truth answer $Y$ with respect to an observer's posterior; by the chain rule for mutual information, we have $H(Y \mid q, b_{1:t}) \leq H(Y \mid q, b_{1:t-1})$ for all $t$, where $b_{1:t} = (b_1, \ldots, b_t)$.

**Overall Dynamics.** We summarize the per-round behavior as a coverage–identification factorization conditioned on the public context $(q, b_{t-1})$. The system first attains *coverage* $p_t$ when the slate contains at least one correct candidate, then achieves *identification* $q_t$ when the central policy selects

a correct candidate. In Appendix B, we show the following cumulative reliability inequality: if we have that both $p_t \geq \underline{p}$ and $q_t \geq \underline{q}$ almost surely for all $t$, then $\Pr(\text{success within } R \text{ rounds}) \geq 1 - (1 - \underline{pq})^R$. An immediate consequence is the following tail inequality: if $R \geq \frac{1}{\underline{pq}} \log\left(\frac{1}{\delta}\right)$, then the probability of success within the first $R$ rounds is at least $1 - \delta$.

### 3.3 CONDITIONAL LISTWISE POLICY OPTIMIZATION (CLPO)

Having established the MAESTRO paradigm, the key question becomes how to *optimize* the convergence process so that the Central Agent can reliably approximate the Bayesian decision rule.[1] Conceptually, the two phases of MAESTRO naturally align with the classical exploration–exploitation trade-off: Phase 1 (divergence) expands the hypothesis space through exploration, while Phase 2 (convergence) serves as exploitation, transforming the diverse candidate set into a single endorsed solution with a supporting rationale. This perspective makes the Phase 2 convergence step a natural fit for reinforcement learning (RL) on the objective:

$$\max_{\pi_\theta} \; \mathbb{E}_{(q, \mathcal{C})}\Big[ r\big(q, \mathcal{C}, \text{Chosen}, \text{Reason}\big) \Big], \tag{5}$$

where $\pi_\theta$ denotes the policy of the Central Agent, $\mathcal{C}$ is the candidate pool of responses sub-sampled over all $R$ rounds, and $r$ is our unified reward, which comprises answer correctness and rationale quality assessed via reasoning attributes, as detailed in Appendix C.1. This formulation highlights the two challenge at the heart of convergence: the Central Agent must both *(i)* provide a coherent and discriminative rationale that distinguishes the endorsed solution from its competitors, and also *(ii)* select the correct decision token.

*Limitations of Naïve Sequence-Level Optimization.* A natural baseline for training (5) is *Group Relative Policy Optimization (GRPO)* (Shao et al., 2024), which contrasts candidates within a group and scales sequence log-probabilities by relative advantage. Although suitable for "pick-one-from-$K$" settings, applying GRPO to full sequences exposes three core issues. First, it behaves like *pointwise supervision*: updates treat each completion in isolation rather than judging rationales by their strength relative to alternatives. Second, the reward signal is *entangled* across decision and rationale tokens, which obscures credit assignment. Third, this entanglement induces *spurious style effects*, where verbosity or lexical patterns receive undue credit and concise reasoning is penalized.

**Conditional Listwise Policy Optimization (CLPO).** We propose CLPO, a decoupled training loss that first learns to produce reliable, discriminative rationales and then learns to make a reliable discrete choice. Concretely, CLPO optimizes the rationale span with a conditional listwise ranking objective (Xia et al., 2008) over the entire candidate set. CLPO allocates the reinforcement signal to the decision tokens, using a focused policy-gradient update to sharpen identification without interference from explanation style. By matching each subproblem to the right objective, CLPO resolves credit entanglement, reduces confounding factors from length/style, and stabilizes training.

*Strategic Decision Loss ($\mathcal{L}_{choice}$).* The convergence phase ultimately hinges on the central agent's ability to make a precise strategic decision: which candidate to endorse. To ensure a clean credit signal, we allocate the reinforcement gradient exclusively to the decision tokens (the choice and its corresponding answer), conditioned on the rationale context. This disentanglement prevents reasoning length or style from interfering with the discrete choice. Formally, we define:

$$\mathcal{L}_{\text{choice}} = -\mathbb{E}\Big[ \sum_{k=1}^{|\mathcal{C}|} A_k \cdot \log \pi_\theta(k \mid q, \mathcal{C}) \Big], \tag{6}$$

where the advantage $A_k = r(c_k) - \bar{r}$, and $\bar{r}$ is the average reward within the candidate set. In practice, we mask the rationale tokens and aggregate log-probabilities only over the decision segment.

*Tactical Argumentation Loss ($\mathcal{L}_{reason\_rank}$).* Agents articulate a justification along with a discrete choice. In our framework, the rationale is generated *before* the final endorsement. We posit that a justification should not only be plausible on its own, but its plausibility should surpass alternatives. To capture this comparative quality, we employ a *Listwise Ranking Loss* (Xia et al., 2008). Formally, let $\sigma$ be the permutation that sorts the rewards in descending order, $r(c_{\sigma_1}) \geq \cdots \geq r(c_{\sigma_{|\mathcal{C}|}})$. Write

---

[1]We do not consider optimizing the policies of the execution agents, which may further improve MAESTRO.

| Type | Mech | Model | GSM8K | MATH | AIME | AMC | MMLU | HumanEval |
|------|------|-------|-------|------|------|-----|------|-----------|
| SA | Ref | Vanilla | 0.7276 | 0.4285 | 0.0296 | 0.0803 | 0.5799 | 0.4756 |
| SA | Ref | CoT | 0.7422 | 0.4693 | 0.0370 | 0.1165 | 0.6157 | 0.5142 |
| SA | Ref | SC | 0.8079 | 0.5128 | 0.0407 | 0.1245 | 0.6830 | 0.5752 |
| MA | Prog | PHP | 0.8001 | 0.5371 | 0.0444 | 0.1566 | 0.6846 | 0.5650 |
| MA | Deb | LLM-Debate | 0.8352 | 0.5625 | 0.0556 | 0.1928 | 0.6759 | 0.5772 |
| MA | Deb | Group-Debate | 0.8398 | _0.5742_ | 0.0519 | _0.2048_ | _0.6989_ | 0.5793 |
| MA | Dyn | DyLAN | 0.8203 | 0.5532 | 0.0370 | 0.1968 | 0.6685 | 0.6159 |
| WF | Dyn | GPTSwarm | _0.8489_ | 0.5669 | _0.0578_ | 0.1566 | 0.6967 | 0.5955 |
| WF | Dyn | AgentPrune | 0.8438 | 0.5437 | 0.0481 | 0.1647 | 0.6909 | 0.5711 |
| WF | Dyn | AFlow | 0.8375 | 0.5528 | 0.0444 | 0.1205 | 0.6931 | _0.6220_ |
| WF | E-S | MAESTRO | 0.8703 | 0.5916 | 0.0556 | 0.2371 | 0.7052 | 0.6267 |
| WF | E-S | w/ SFT | 0.8769 | 0.5983 | 0.0538 | 0.2482 | 0.7085 | 0.6321 |
| WF | E-S | w/ GRPO | 0.8867 | 0.6129 | 0.0704 | 0.2630 | 0.7168 | 0.6538 |
| WF | E-S | w/ CLPO | **0.8933** | **0.6285** | **0.0851** | **0.2852** | **0.7238** | **0.6687** |

Table 1: Comparison of baseline and proposed methods using the LLaMA-8B backbone. The table organizes models by Type (SA: single-agent, MA: multi-agent, WF: workflow-style framework) and by Mechanism (Reflection, Progressive Prompting, Debate, Dynamic Coordination, and Exploration–Synthesis). Underlined numbers indicate the best-performing baseline on each benchmark. Additional results with two recent RL-based baselines (ReMA and MAS-GPT) are reported in Appendix D (Table 17), and results on the GPQA scientific QA benchmark are given in Table 18.

the justification for the $k$-th candidate in $\mathcal{C}$ as a token sequence $y_{k,1:L_k}$. We define this loss as:

$$\mathcal{L}_{\text{reason\_rank}} = -\sum_{j=1}^{|\mathcal{C}|} \log \frac{\exp(s_{\sigma_j})}{\sum_{l=j}^{|\mathcal{C}|} \exp(s_{\sigma_l})}, \quad s_k = \frac{1}{L_k} \sum_{\tau=1}^{L_k} \log \pi_\theta\big(y_{k,\tau} \mid y_{k,1:\tau-1},\ q,\ \mathcal{C}\big). \quad (7)$$

**The CLPO Objective.** The CLPO training objective combines the two losses (6) and (7), with standard regularization terms to ensure stable exploration and prevent catastrophic forgetting. The policy is regularized towards a reference policy $\pi_{\text{ref}}$ (e.g., the initial SFT model) via a KL-divergence term $\mathcal{L}_{\text{KL}} = \mathbb{E}\left[D_{\text{KL}}(\pi_\theta(\cdot \mid q, \mathcal{C}) \,\|\, \pi_{\text{ref}}(\cdot \mid q, \mathcal{C}))\right]$ and an entropy bonus $\mathcal{L}_{\text{Entropy}} = \mathbb{E}\left[H(\pi_\theta(\cdot \mid q, \mathcal{C}))\right]$ that encourages exploration of justifications. The final objective is:

$$\mathcal{L}_{\text{CLPO}} = \mathcal{L}_{\text{choice}} + \lambda_{\text{rank}} \cdot \mathcal{L}_{\text{reason\_rank}} + \lambda_{\text{kl}} \cdot \mathcal{L}_{\text{KL}} - \lambda_{\text{ent}} \cdot \mathcal{L}_{\text{Entropy}}. \quad (8)$$

As we will see shortly, by decoupling the learning objectives for strategic choice and tactical argumentation, CLPO delivers a richer and more stable gradient signal, ensuring clean credit assignment.

# 4 EXPERIMENTS

**Experimental Setup.** We evaluate our approach across diverse benchmarks, including mathematical reasoning (GSM8K, MATH, AIME, AMC), factual and analytical reasoning (MMLU), and program synthesis (HumanEval), using Solve Rate, Accuracy, and Pass@1 as evaluation metrics. Baselines span single-agent reasoning methods, peer-interaction frameworks, routing and topology controllers, workflow and graph search approaches, and communication-efficient systems. Unless otherwise noted, experiments use three agents and three communication rounds, with instruction-tuned LLaMA-3B/8B and Qwen-3B/7B models under standard nucleus sampling. All reported results are averaged over three random seeds. See Appendix C.1 for a full account of settings.

## 4.1 MAIN EXPERIMENTS

**Overall Performance.** Table 1 shows that MAESTRO consistently surpasses both single-agent and multi-agent baselines across six reasoning benchmarks. On the trainable backbone LLaMA-8B, MAESTRO with CLPO achieves state-of-the-art accuracy, reaching 89.33% on GSM8K and 28.52% on AMC, which corresponds to average gains of 4%–8% over strong baselines such as GPTSwarm, AgentPrune, and Group-Debate. The improvements arise from two complementary effects: parallel

| Dataset | Vanilla | CoT | SC | Debate | GPTS | AP | AF | MAESTRO |
|---------|---------|-----|-----|--------|------|------|------|---------|
| GSM8K | 93.17 | 93.68 | 93.32 | 94.66 | 94.66 | 94.89 | 92.30 | **95.60** |
| MMLU | 77.81 | 78.43 | 81.05 | 81.04 | 82.80 | 83.02 | 83.10 | **84.09** |
| HumanEval | 85.71 | 86.69 | 87.58 | 84.38 | 86.28 | 86.80 | 90.06 | **90.65** |

Table 2: Performance comparison on GSM8K, MMLU, and HumanEval using a GPT-4o-mini backbone. MAESTRO consistently achieves the highest accuracy across all benchmarks, outperforming both single-agent methods (Vanilla, CoT, and SC) and existing multi-agent frameworks like Debate, GPTSwarm (GPTS), AgentPrune (AP), and AFlow (AF).

| Model | LLaMA-8B | LLaMA-3B | Qwen-7B | Qwen-3B |
|-------|----------|----------|---------|---------|
| Vanilla | 0.7276 | 0.4685 | 0.9088 | 0.8337 |
| CoT | 0.7422 | 0.5014 | 0.9098 | 0.8456 |
| SC | 0.8079 | 0.5421 | 0.9295 | 0.8860 |
| PHP | 0.8001 | 0.6222 | 0.9330 | 0.8645 |
| LLM-Debate | 0.8352 | 0.7584 | 0.9363 | 0.8714 |
| DyLAN | 0.8203 | 0.7647 | 0.9315 | 0.8810 |
| GPTSwarm | 0.8489 | 0.6919 | 0.9227 | 0.8678 |
| AgentPrune | 0.8438 | 0.6502 | 0.9244 | 0.8643 |
| AFlow | 0.8375 | 0.6837 | 0.9286 | 0.8752 |
| MAESTRO w/ CLPO | **0.8933** | **0.8153** | **0.9512** | **0.9083** |

Table 3: Performance of collaborative reasoning baselines across four backbone LLMs (LLaMA-8B, LLaMA-3B, Qwen-7B, Qwen-3B) on GSM8K. MAESTRO w/ CLPO consistently achieves the highest accuracy, demonstrating robustness and generality across model architectures. Additional results with a larger backbone, LLaMA3-70B, are reported in Appendix D (Table 9).

exploration increases coverage, while CLPO strengthens the central selector's ability to identify correct solutions. This dual mechanism is especially beneficial on competition-style math tasks such as AMC and AIME, where incorrect but fluent candidates often mislead majority-voting or self-consistency. Importantly, the gains are not limited to trainable backbones. As shown in Table 2, even with the closed-source GPT-4o-mini under a prompt-only setting, MAESTRO achieves the best or tied-best results on GSM8K, MMLU, and HumanEval. The consistency across open- and closed-source models indicates that improvements stem from the collaborative orchestration itself rather than parameter updates, establishing MAESTRO as a robust paradigm for LLM collaboration.

**Cross-Backbone Consistency.** To further examine the generality of our optimization strategy, we applied MAESTRO with CLPO across different LLM backbones, including LLaMA-8B, LLaMA-3B, Qwen-7B, and Qwen-3B (Table 3). We observe consistent improvements across all settings. On GSM8K, the accuracy reaches 89.33% with LLaMA-8B, 81.53% with LLaMA-3B, 95.12% with Qwen-7B, and 90.83% with Qwen-3B, establishing clear gains compared to their strongest respective baselines. The effectiveness of CLPO is not confined to a specific model family or size. Instead, as shown in Table 4, the optimization consistently enhances the identification probability $q_t$, enabling the central synthesis agent to more reliably distinguish correct solutions from plausible distractors. Importantly, this pattern is also reflected on AMC, where the improvements are similarly pronounced, underscoring that the collaborative mechanism combined with CLPO is broadly transferable across architectures. Overall, these findings confirm that MAESTRO with CLPO achieves robust gains across backbones, validating the universality of our collaborative optimization paradigm.

## 4.2 ANALYSIS EXPERIMENTS

**Centralized Paradigm Variants: Selection vs. Generation.** We now examine two natural centralized paradigms for convergence, namely generation and selection, to clarify why the latter forms the core of MAESTRO (Figure 2). When the central agent directly generates a reasoning trajectory and final answer (CENTRAL-GEN), the accuracy drops substantially. Incorporating self-consistency into generation (CENTRAL-GEN+SC) yields only a marginal gain. In contrast, the selection paradigm (CENTRAL-SELECT, ours) described in Section 3.2 attains the highest accuracy, reaching 0.870 on

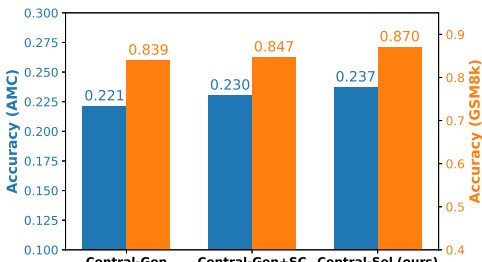 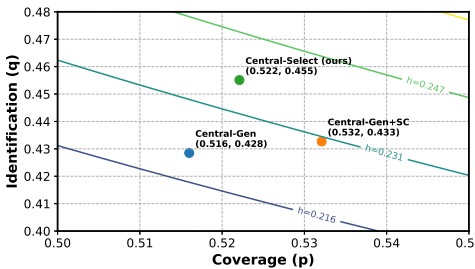

Figure 2: Comparison of collaboration paradigms. **Left:** task accuracy on AMC and GSM8K across different central coordination strategies. **Right:** performance decomposed into coverage and identification rates; central selection transforms diverse reasoning into reliable outcomes.

GSM8K and 0.237 on AMC. Figure 2 (Right) further decomposes the results into coverage and identification probabilities. The three paradigms exhibit similar coverage, but the identification probability is markedly higher under CENTRAL-SELECT (0.8919 on GSM8K and 0.4551 on AMC), which directly explains its superior end-to-end accuracy. We hypothesize that generation forces the central agent to absorb long and noisy contexts from multiple candidates, often diluting critical distinctions and amplifying misleading patterns. LLMs also tend to prioritize narrative coherence over factual correctness, which makes direct generation vulnerable to self-consistent hallucinations (Farquhar et al., 2024; Banerjee et al., 2025). Self-consistency mitigates randomness but cannot overcome these structural issues. By contrast, the selection paradigm frames convergence as a discriminative comparison among competing candidates, thereby preserving informative differences and reliably elevating correct solutions. Figure 5 (appendix) contains a complementary visualization.

**Evidence versus Verdict in Centralized Selection.** We examine how different types of candidate information influence the central selector's decisions by comparing three settings: *Reason-only* (only reasoning steps), *Answer-only* (only the final answer), and *Both* (ours, reasoning with the answer). As shown in Figure 3, *Answer-only* yields the weakest performance (GSM8K 0.840, AMC 0.205), while *Reason-only* performs better but remains slightly below the full setting. Since the candidate pool is identical, coverage is unchanged and differences arise from identification capability. The results show that reasoning and outcomes play complementary roles. Without reasoning, the selector lacks evidential structure and often falls back on superficial heuristics. Without final outcomes, it struggles to resolve cases where plausible reasoning paths diverge to different answers. Combining both provides the strongest performance: reasoning paths supply discriminative evidence, while answers anchor the verdict and disambiguate close cases.

**Disentangled Optimization Signals in CLPO.** To better understand the contribution of each optimization signal in CLPO, we conduct an ablation study by removing either the decision-focused loss $\mathcal{L}_{\text{choice}}$ or the rationale ranking loss $\mathcal{L}_{\text{reason}}$ (Figure 3). Removing $\mathcal{L}_{\text{choice}}$ causes a modest decline, indicating that ranking-based supervision over rationales alone can sustain reasonable convergence. In contrast, removing $\mathcal{L}_{\text{reason}}$ leads to a sharp degradation (AMC 0.261, GSM8K 0.881); without comparative evaluation of explanations, the selector is more easily swayed by persuasive but incorrect candidates. The full CLPO objective achieves the best performance, confirming the necessity of combining both terms. This pattern aligns with our design intuition: $\mathcal{L}_{\text{choice}}$ strengthens decisiveness by refining the probability of endorsing the correct candidate, while $\mathcal{L}_{\text{reason}}$ enforces discriminative evidence quality by forcing correct rationales to outrank distractors. Their joint effect provides clean credit assignment across decisions and justifications, ensuring that convergence is accurate.

### 4.3 HYPER-PARAMETER ANALYSIS

**Scaling Agent Populations and Collaboration Rounds.** We study how the size of the agent collective and the number of collaboration rounds influence performance. As shown in Figure 4, increasing the population from two to four steadily improves accuracy (AMC $0.253 \rightarrow 0.3052$; GSM8K $0.8693 \rightarrow 0.9037$). Broader exploration raises the probability that at least one candidate is correct, and the central selector trained with CLPO can convert this coverage into higher identification accuracy. Beyond four agents, however, gains saturate and slightly decline because redundancy introduces distractors. A similar trend appears when varying the number of collaboration rounds (see

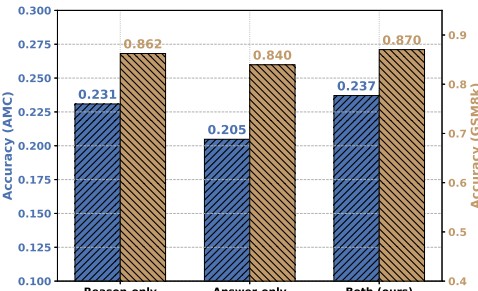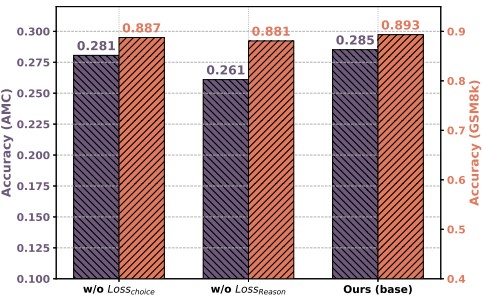

Figure 3: Ablation studies on central selection inputs and CLPO losses. **Left:** Reason-only, Answer-only, and Both settings when passing candidate information to the central selector. **Right:** contributions of loss components studied by removing choice or reasoning supervision.

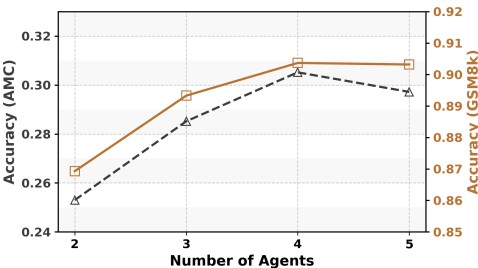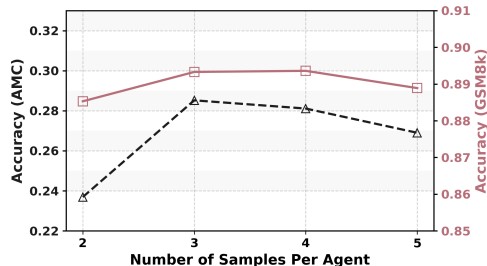

Figure 4: Effect of collaborative scale on reasoning performance. **Left:** accuracy with varying agent numbers. **Right:** impact of sampling multiplicity. In each, the solid line corresponds to GSM8K accuracy (right axis) and the dashed line corresponds to AMC accuracy (left axis).

Figure 6 in the appendix). Adding one or two rounds improves identification by focusing exploration through broadcasted evidence, while coverage changes little once the initial pool is large. Excessive rounds reduce diversity, amplify early errors through herding, and increase stochastic variance. The results highlight a consistent trade-off: additional agents and rounds enhance coverage and identification up to a point, but beyond that redundancy and bias dominate. Moderate settings of 3–4 agents and 2–3 rounds achieve the best balance. A more detailed analysis is provided in Appendix C.2.

**Sampling Depth per Agent: Coverage–Variance Trade-off.** We examine how the number of samples per agent ($K$) affects performance (Figure 4). As $K$ increases from 2 to 3, accuracy improves (AMC $0.2369 \rightarrow 0.2852$; GSM8K $0.8853 \rightarrow 0.8933$). After, gains saturate: GSM8K changes little at $K=4$ and declines at $K=5$, while AMC peaks at $K=3$ before dropping, reflecting the exploration–synthesis decomposition. Larger $K$ initially raises coverage, but multi-sampling from the same policy quickly yields correlated and redundant outputs; deeper sampling inflates within-round variance by drawing repeatedly from one agent rather than diversifying across agents. Once coverage nears saturation, additional samples contribute more noise than signal. These results suggest a practical guideline: allocate budget to enlarging the number of agents to diversify hypotheses, while keeping $K$ modest so that the selector can reliably convert coverage into identification. A more detailed analysis is provided in Appendix C.2.

## 5 CONCLUSION

We introduce MAESTRO, a principled framework for multi-agent collaboration that enables both divergent exploration and convergent synthesis. We also present CLPO, an RL method that achieves precise credit assignment through decision-focused optimization and comparative supervision. Together these components yield consistent improvements across diverse reasoning benchmarks and surpass state-of-the-art multi-agent methods. Looking ahead, we plan to investigate unified policy objectives that jointly optimize exploration and synthesis, and continuous learning paradigms that enable multi-agent collectives to refine collaboration dynamics through self-improvement over time.

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

# A A COMPREHENSIVE REVIEW OF RELATED WORK

## A.1 MULTI-AGENT LLM COLLABORATION

Single LLM agents, despite their impressive individual capabilities, face fundamental limitations in context length, sequential generation, and breadth of expertise. These constraints hinder performance on tasks that demand parallel information processing, complementary skill sets, and the synthesis of diverse perspectives (Gabriel et al., 2024; Liang et al., 2023; Xiong et al., 2023; Yin et al., 2023; Zhang et al., 2023). To overcome these bottlenecks, researchers have increasingly turned to multi-agent systems (MAS), where collectives of LLM-powered agents coordinate to realize forms of collective intelligence in domains such as software engineering, complex planning, and scientific discovery (Hong et al., 2023; Chen et al., 2023b; Jiang et al., 2023; Ning et al., 2023; Qiao et al., 2024; Pan et al., 2024; Suzgun & Kalai, 2024; Chen et al., 2023a; Ishibashi & Nishimura, 2024).

Early approaches largely follow a *prompt-based paradigm*, where roles, protocols, and workflows are specified by hand. Debate-style and critique frameworks (Du et al., 2023; Chan et al., 2023; Chen et al., 2024; Mukobi et al., 2023; Wang et al., 2023; Abdelnabi et al., 2024) as well as corporate-style pipelines such as MetaGPT (Hong et al., 2023; Qian et al., 2023) exemplify this direction. These systems demonstrate the promise of structured collaboration but remain brittle because their strategies are statically prescribed and cannot adapt or learn from experience (Jiang et al., 2023; Liang et al., 2023; He et al., 2023).

Beyond static prompting, recent work introduces more principled coordination schemes. *Prestructured paradigms* adopt fixed interaction topologies, such as chains, trees, or graphs, to organize communication and enforce critique (Du et al., 2023; Liu et al., 2024; Qian et al., 2024). In parallel, *self-organizing paradigms* dynamically adapt collaboration graphs during inference using search, pruning, or routing methods, as seen in DyLAN, MasRouter, GPTSwarm, and AFLOW (Liu et al., 2023b; Hu et al., 2024b; Shang et al., 2024; Zhang et al., 2024b; Zhuge et al., 2024; Zhang et al., 2024c; Hu et al., 2024a; Yue et al., 2025). These frameworks improve efficiency and flexibility, yet they often reduce coordination to architectural wiring and lack mechanisms for fine-grained credit assignment.

Complementary efforts focus on *role specialization* and *organizational analogies*, where agents are differentiated as planners, solvers, or verifiers, or even structured as corporate roles such as CEO and engineer (Hong et al., 2023; Li et al., 2023; Mandi et al., 2024; Talebirad & Nadiri, 2023; Du et al., 2023; Chen et al., 2023b). Communication protocols vary between centralized, decentralized, and hierarchical settings, as well as synchronous versus asynchronous exchanges (Jiang et al., 2023; Ning et al., 2023; Pan et al., 2024; Liang et al., 2023; Zhang et al., 2023; Du et al., 2023; Chan et al., 2023; Chen et al., 2024). These design choices trade off scalability, robustness, and overhead but leave unresolved the fundamental question of how to separate decision-making from justification in a principled manner.

Overall, existing MAS paradigms illuminate diverse strategies for orchestrating collaboration, but they remain limited in their ability to balance broad exploration with reliable convergence and to assign credit cleanly across agents and rationales.

## A.2 REINFORCEMENT LEARNING FOR MULTI-AGENT LLMS.

A central trend in multi-agent LLM research is to move beyond static prompt engineering toward learning from interaction. Early work explored supervised fine-tuning (SFT) on expert demonstrations, which injects cooperative behaviors by imitation but is limited in adaptability to unseen coordination settings (Madaan et al., 2023; Zelikman et al., 2024). In contrast, reinforcement learning (RL) supplies a reward-driven mechanism that allows agents to refine strategies from experience and discover emergent collaboration patterns (Zhu et al., 2025; Zhuang et al., 2024). In practice, SFT often initializes base policies, while multi-agent reinforcement learning (MARL) further tailors them under task feedback (Zhu et al., 2025; Li, 2019; Zhang et al., 2021).

Recent efforts fall into three complementary directions. First, some approaches compile language into structured controllers before learning, such as translating dialogue into plans, graphs, or code, which grounds RL optimization in compact symbolic spaces (Zhuang et al., 2024; Jia et al., 2025). Second, others focus on adaptive collaboration online, dynamically refining task decomposition,

agent assignment, or communication routing through RL signals (Zhou et al., 2025; Wang et al., 2024; Xu et al., 2025; Li et al., 2024a). Third, direct policy optimization for reasoning behaviors has gained traction, with GRPO- and PPO-style updates applied to cooperative justification and answer selection, often combined with tool use or human feedback (Wan et al., 2025; Park et al., 2025; Han et al., 2025). Across these directions, RL provides the flexibility to align multi-agent dynamics with task objectives rather than relying solely on fixed prompts or wiring rules.

At the same time, this line of work highlights several core challenges. A prominent difficulty is credit assignment: linguistic outputs entangle the correctness of discrete decisions with the plausibility of accompanying rationales, making it unclear what aspect of behavior is being rewarded (Wei et al., 2025; Jiang et al., 2025). Another challenge is efficient exploration in vast language action spaces, where agents may generate superficially diverse but semantically redundant outputs (Liu et al., 2023a; Zhang et al., 2024d). Finally, there is the issue of alignment of emergent behaviors, since collaboration can amplify biases or drift without proper reward shaping (Alsadat & Xu, 2024; Lin et al., 2025).

Our work follows this trajectory while placing a sharper emphasis on the convergence step of collaboration. Rather than treating group outcomes as a monolithic reward signal, we recast convergence as a structured optimization problem that separates the supervision of rationales from decision signals. This perspective motivates the design of a new RL objective that provides comparative supervision across rationales while preserving clean decision gradients, complementing existing GRPO-style multi-agent optimization.

## B    DERIVATION OF THE CUMULATIVE RELIABILITY INEQUALITY

In this section we provide a derivation of the cumulative reliability inequality from Section 3.2.

We start by defining the history (i.e., filtration) $\mathcal{F}_t := (q, \theta_{1:t}, \zeta_{1:t})$, with $\mathcal{F}_0 := q$, where $\theta_{1:t}$ denotes the randomness of the execution agents for the first $t$ rounds, and $\zeta_{1:t}$ denotes the randomness of the central agent for the first $t$ rounds. Let $\mathsf{Cand}_t := \{\exists\, (i, k) \in [N] \times [K] \text{ s.t. } E(c_{t,k}^{(i)}) \text{ holds}\}$ denote the event the round $t$ slate $\mathcal{C}_t$ contains at least one correct candidate. Let $(i_t, k_t) \sim \pi_\theta(\cdot \mid q, \mathcal{C}_t)$ be the central decision made at time $t$. Then we have that, assuming $p_t \geq \underline{p}$ and $q_t \geq \underline{q}$ for non-random $\underline{p}, \underline{q}$ almost surely for all $t$:

$$
\begin{aligned}
h_t &:= \Pr(\text{Success}_t \mid \mathcal{F}_{t-1}) \\
&= \mathbb{E}_{\mathcal{C}_t \mid \mathcal{F}_{t-1}}[\Pr(\text{Success}_t \mid \mathcal{F}_{t-1}, \mathcal{C}_t)] \\
&\overset{(a)}{=} \mathbb{E}_{\mathcal{C}_t \mid \mathcal{F}_{t-1}}[\mathbf{1}\{\mathsf{Cand}_t\} \Pr((i_t, k_t) \in S_t \mid q, \mathcal{C}_t, \{|S_t| \geq 1\})] \\
&\overset{(b)}{=} \mathbb{E}_{\mathcal{C}_t \mid \mathcal{F}_{t-1}}[\mathbf{1}\{\mathsf{Cand}_t\} q_t] \\
&\overset{(c)}{\geq} \mathbb{E}_{\mathcal{C}_t \mid \mathcal{F}_{t-1}}[\mathbf{1}\{\mathsf{Cand}_t\}]\underline{q} \\
&= \Pr(\mathsf{Cand}_t \mid \mathcal{F}_{t-1})\underline{q} \\
&\overset{(d)}{=} \Pr(\mathsf{Cand}_t \mid q, s_t^{(1:N)})\underline{q} \\
&\overset{(e)}{=} p_t \underline{q} \\
&\overset{(f)}{\geq} \underline{p}\underline{q},
\end{aligned}
$$

where in (a) we used the fact that the decision $(i_t, k_t)$ is generated conditioned only on $(q, \mathcal{C}_t)$, (b) is the definition of $q_t$ from (4), (c) uses our lower bound assumption on $q_t$, (d) uses the fact that the candidate decisions $\mathcal{C}_t$ are generated conditioned on $(q, z_{t-1}^{(1:N)}, b_{t-1})$, which is contained within $\mathcal{F}_{t-1}$, (e) is the definition of $p_t$ from (2), and (f) uses our lower bound assumption on $p_t$.

Now, let $X_t := \mathbf{1}\{\text{Success}_t\}$. By definition we have that $X_t$ is $\mathcal{F}_t$-measurable. Hence by repeated applications of the tower-property of conditional expectations,

$$
\begin{aligned}
\Pr(\text{Fail all } R \text{ rounds}) &= \mathbb{E}\left[\prod_{t=1}^{R}(1-X_t)\right] \\
&= \mathbb{E}\left[\mathbb{E}\left[\prod_{t=1}^{R}(1-X_t) \mid \mathcal{F}_{R-1}\right]\right] \\
&= \mathbb{E}\left[\prod_{t=1}^{R-1}(1-X_t)\mathbb{E}\left[1-X_R \mid \mathcal{F}_{R-1}\right]\right] \\
&= \mathbb{E}\left[\prod_{t=1}^{R-1}(1-X_t)(1-h_R)\right] \\
&\overset{(a)}{\leq} \mathbb{E}\left[\prod_{t=1}^{R-1}(1-X_t)\right](1-\underline{pq}) \leq \cdots \leq (1-\underline{pq})^R,
\end{aligned}
$$

where (a) follows from above where we established $h_t \geq \underline{pq}$.

## C   EXPERIMENT

### C.1   EXPERIMENTAL SETTINGS

**Datasets & Benchmarks.**   We evaluate the framework on three task families designed to stress complementary aspects of collective reasoning, namely precise numeric inference, broad factual and analytical judgment, and executable synthesis. This spectrum assesses both the "diverge" capacity (hypothesis coverage) and the "converge" capacity (principled selection).

*Mathematical reasoning.* We use **GSM8K**, **MATH**, **AIME**, and **AMC**. GSM8K comprises grade-school word problems with single numeric targets; MATH covers competition-level problems across algebra, number theory, geometry, and combinatorics; AIME consists of short-answer olympiad items with integer solutions; AMC includes large-scale contest questions (we report on the standard subset with unambiguous numeric targets). Performance is measured by *Solve Rate*, the proportion of items whose predicted answer exactly matches the ground truth under benchmark normalization rules.

*General reasoning.* We use **MMLU**, spanning 57 subjects from STEM to humanities under a four-choice multiple-choice format. Performance is reported as *Accuracy*, i.e., the fraction of correctly selected options, under the benchmark's standard few-shot setting.

*Code generation.* We use **HumanEval**, where models synthesize functions from natural-language specifications. Performance is reported as *Pass@1*, the percentage of prompts for which the single generated solution passes all hidden unit tests.

Unless otherwise noted, we follow official splits and prompting guidelines, do not use external tools or retrieval, and keep evaluation deterministic for single predictions. When stochastic sampling is required (e.g., for self-consistency or multi-agent generation), we fix seeds and average over repeated runs; confidence intervals are reported in the appendix. This protocol ensures comparability with prior work while isolating the contribution of the collaboration paradigm and training objective.

**Baselines.**   We compare against collaborative LLM methods organized by their underlying *collaboration mechanism*, rather than model brand. (i) *Single-agent reasoning*: **Vanilla** (direct decoding), **CoT** (chain-of-thought prompting), and **SC** (self-consistency with majority vote). (ii) *Peer interaction*: **LLM-Debate** (multi-round argumentation with shared transcripts), **GroupDebate** (multi-agent debate with voting-based aggregation) and **PHP** (pairwise critique without a global selector). (iii) *Routing/topology control*: **DyLAN** (layered agent network with pruning and early-stop consensus). (iv) *Workflow/graph search*: **GPTSwarm** (optimization of reasoning graphs over multiple prompting strategies) and **AFLOW** (Monte-Carlo search over reusable operators). (v) *Communication efficiency*: **AgentPrune** (sparse message passing to reduce cost while maintaining accuracy).

For all baselines we use the same base models, adopt each method's official prompts and stopping criteria, and match collaboration budgets (rounds, agents, and generations). When methods output multiple candidates, we apply their canonical aggregation (e.g., majority vote or ranker). This taxonomy clarifies whether improvements come from stronger generation (divergence), more reliable selection (convergence), or better workflow, providing a diagnostic comparison to our exploration–synthesis paradigm.

**Prompt Templates.** To make our experimental setup transparent and reproducible, we explicitly document the instruction prompts used by different agents in our framework. These templates capture the roles and responsibilities of both reasoning agents and the center arbiter, highlighting how they collaborate through structured interaction. For clarity, we present concrete examples in the domain of mathematical reasoning problems, which serve as a representative case for illustrating the prompt design.

---

**Execution Agent Prompt (Initial Round)**

You are Reasoning Agent #{agent_id}. Your task is to carefully solve the given math problem step by step. Clearly show your reasoning process, making sure that each transformation is logically valid. Avoid skipping important intermediate steps.
At the end of your reasoning, provide the final numeric answer in the exact format: `\boxed{...}`.
**Problem:** {{math_question}}

---

**Execution Agent Prompt (Interactive Round)**

You are Reasoning Agent #{agent_id}. You previously proposed multiple solutions and now also receive the Center Arbiter's synthesis.
Re-evaluate the problem carefully, considering both your earlier solutions and the Arbiter's feedback. Generate refined solutions that correct any mistakes if needed, ensuring logical consistency.
Each output must end with the final numeric answer in the exact format: `\boxed{...}`.
**Problem:** {{math_question}}
**Your Previous Solutions:** {...}
**Center Arbiter's Feedback:** {...}

---

**Central Agent Prompt**

You are the Center Arbiter, responsible for evaluating candidate solutions proposed by agents. Carefully read the original problem and all candidate solutions. Compare their reasoning, detect mistakes if present, and identify the most reliable candidate.
Then, following the strict format below, provide a short justification, the chosen candidate index, and the final numeric answer in `\boxed{...}`.
**Problem:** {{math_question}}
**Candidates:**
- Candidate 1: {...}
- Candidate 2: {...}
- Candidate 3: {...}

**STRICT OUTPUT FORMAT:**
Reason: {detailed justification}
Chosen: {candidate_id}
Final: `\boxed{...}`

---

**Implementation Details.** Our experiments are conducted with a compact configuration where three agents interact across three communication rounds for each query. The agents are instantiated from widely used instruction-tuned models including **Llama-3.1-8B-Instruct**, **Llama-3.2-3B-**

**Instruct** (Dubey et al., 2024), and **Qwen2.5-7B-Instruct** as well as **Qwen2.5-3B-Instruct** (Team, 2024). All models are accessed through the HuggingFace Transformers library with 8-bit quantization to reduce GPU memory usage. We enable KV caching throughout the experiments to improve generation efficiency. We adopt a unified generation setup across all experiments. Unless otherwise noted, nucleus sampling with $p = 0.95$ is used and the maximum output length is set to $512$ tokens. The default temperature is $0.7$, which balances diversity and stability. For tasks requiring deterministic evaluation, such as pairwise preference comparisons or revision prompts, we reduce the temperature to $0.3$. The central agent is always assigned a temperature of $0.0$ to enforce deterministic decisions and avoid stochastic drift. To ensure comparability across methods, all models share the same decoding settings and random seeds are fixed. This setup follows common practice in LLM evaluation and ensures that performance differences stem from the collaboration paradigm rather than decoding hyperparameters. During both supervised fine-tuning (SFT) and policy optimization, we adopt parameter-efficient fine-tuning using LoRA. Unless otherwise noted, the LoRA rank is set to $16$, with scaling factor $\alpha = 32$ and dropout $0.05$. Only LoRA parameters, LayerNorm statistics, and bias terms are updated, while all other weights remain frozen. Training uses Adam ($\beta_1 = 0.9$, $\beta_2 = 0.999$, $\epsilon = 10^{-8}$) with an initial learning rate of $5 \times 10^{-5}$, decayed following a cosine schedule. Gradient norms are clipped at $1.0$ to stabilize optimization. We train with a global batch size of $256$ distributed across four A100 GPUs (80GB each), using mixed precision (bfloat16) for efficiency. The rank-loss coefficient is tuned over $\{0.1, 0.5, 0.8, 1.0\}$. To encourage exploration we add an entropy bonus of $0.01$, while maintaining consistency with the SFT reference policy via a KL regularization weight of $0.1$. Each run proceeds for three epochs, and results are averaged over three random seeds (25, 42, and 99) to ensure robustness and mitigate variance.

**Unified Reward.** We define unified reward that combines answer correctness and reason quality. For each candidate $c_i$, the correctness score is $\mathrm{acc}_i \in [0, 1]$, defined as $\mathrm{acc}_i = \mathbf{1}[\mathrm{answer}_i = \hat{a}]$ for objective/numeric tasks (with an optional tolerance) or as a test-pass rate for programming tasks. Reason quality is computed from the candidate's rationale (and its own final answer/interface statement) via a unified set of binary attributes: structure/readability (stepwise clarity, explicit intermediate quantities), soundness to own answer (derivation strictly leads to its own final answer without contradictions), constraint/format adherence (range, integrality, lowest terms, function signature, etc.), premise/evidence alignment (key facts in the rationale match the prompt or given materials), error diagnosis/refutation (identifies common failure modes or flaws in competing candidates and explains their mechanism), and executability/safety (for implementation tasks, the rationale is consistent with runnability/safety without unjustified risky operations). Each attribute is recognized jointly by programmatic checks (rules, AST/signature validation, equation re-evaluation, range/unit checks, keyword/pattern matching) and a GPT judge that performs semantic recognition over the question and candidate and outputs {Yes, No}. Denoting the decision on attribute $k$ as $d_k(c_i) \in \{1, 0\}$, we average only determined attributes to obtain a single rationale score $s_i = \frac{1}{|\mathcal{V}_i|} \sum_{k \in \mathcal{V}_i} d_k(c_i)$, $\mathcal{V}_i = \{ k \mid d_k(c_i) \in \{0, 1\} \}$. The training reward is a simple weighted fusion, $R_i = w_c \, \mathrm{acc}_i + w_r \, s_i$, $w_c, w_r \geq 0$, $w_c + w_r = 1$, with the practical choice $w_c \geq w_r$ to discourage "fluent but wrong" rationales. Considering the trade-off between prioritizing correctness and still learning discriminative rationales, we adopt the simple default weights $w_c = 0.6$ and $w_r = 0.4$. And we employ gpt-4o as the semantic judge for attribute recognition.

## C.2 EXPERIMENTAL RESULTS.

**Number of Rounds: balancing evidence aggregation and bias amplification.** Figure 6 shows that increasing the number of collaboration rounds improves performance at first, then saturates and may decline. GSM8K peaks around two rounds and AMC around three rounds. This pattern matches our coverage–identification decomposition. The first additional round injects public evidence through broadcast, which focuses subsequent exploration and lifts the identification probability $q_t$ because the central policy compares candidates under a clearer hypothesis space. Coverage $p_t$ changes little once the initial candidate pool is large, so early gains are mainly due to improved identification. Beyond the peak, returns diminish and can turn negative. Repeated conditioning on previous broadcasts reduces effective diversity and increases redundancy, which lowers the probability that new rounds add genuinely novel evidence. If an early broadcast is confidently wrong, later rounds tend to herd toward the same error, creating bias amplification that hurts $q_t$. More rounds also introduce additional stochastic variance while consuming budget, which further limits net gains.

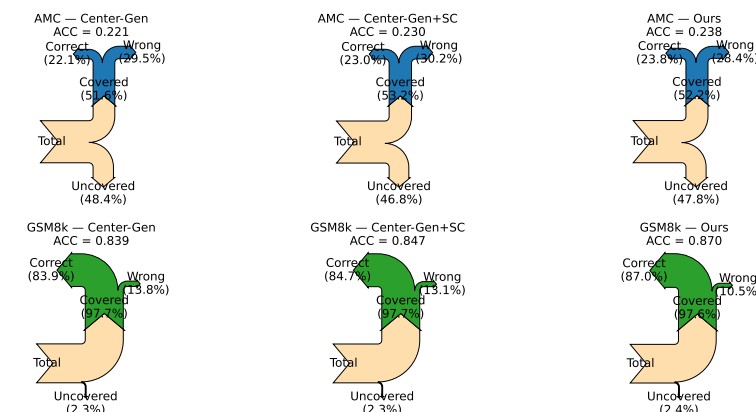

Figure 5: Sankey diagram illustrating performance on AMC and GSM8K under different central coordination strategies. Each flow decomposes accuracy into coverage and identification outcomes, showing how centralized selection more effectively converts diverse reasoning into correct solutions.

| Dataset | Rounds | $p_t$ | $q_t$ | Acc |
|---------|--------|-------|-------|-----|
| GSM8K | 2 | 0.9272 | 0.9479 | 0.8789 |
| | 3 | 0.9325 | **0.9580** | **0.8933** |
| | 4 | **0.9417** | 0.9422 | 0.8872 |
| | 5 | 0.9280 | 0.9528 | 0.8842 |
| AMC | 2 | 0.3395 | 0.8397 | 0.2851 |
| | 3 | 0.3494 | 0.8449 | **0.2952** |
| | 4 | 0.3412 | **0.8477** | 0.2892 |
| | 5 | **0.3503** | 0.8368 | 0.2931 |

Table 4: Comparison of coverage, identification, and accuracy (ACC) on GSM8K and AMC under LLaMA3-8B backbones.

Overall, a small number of rounds is most effective: two rounds on GSM8K and two to three rounds on AMC strike a good balance by converting collective coverage into reliable identification without over-conditioning the agents.

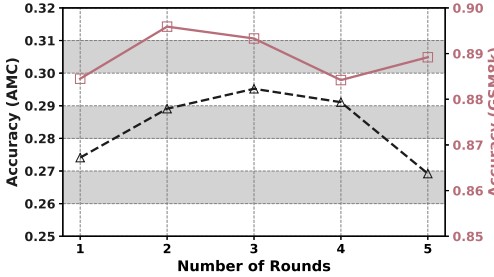

Figure 6: Effect of collaboration rounds on performance. Accuracy is reported for AMC and GSM8K. Performance improves with additional rounds up to a moderate level, then saturates or declines, highlighting the trade-off between evidence aggregation and bias amplification.

**Sampling Depth per Agent: Coverage–Variance Trade-off.** We analyze how the number of samples drawn by each agent ($K$) affects collaborative performance. As shown in Figure 4 (b), increasing $K$ from 2 to 3 improves accuracy on both AMC (from 0.2369 to 0.2852) and GSM8K (from 0.8853 to 0.8933). Beyond this point the gains saturate: GSM8K changes marginally at $K{=}4$ (0.8936) and declines at $K{=}5$ (0.8889), while AMC peaks at $K{=}3$ and then drops to 0.2811 and

0.2690. This pattern reflects our exploration–synthesis decomposition. Increasing $K$ initially raises the chance that at least one candidate is correct, which improves coverage $p_t$. However, multi-sampling from the *same* agent policy quickly becomes correlated and redundant, so the marginal gain in coverage diminishes. At the same time the candidate set grows and introduces more plausible distractors, which elevates the burden on the central selector and can depress identification $q_t$. In practice, deeper per-agent sampling also inflates within-round variance because it relies on stochastic decoding from a single policy instance rather than diversifying across agents. Consequently, once coverage is near saturation, additional $K$ contributes more noise than signal and identification becomes the limiting factor. Taken together with the agent-scaling results, these observations suggest a practical guideline: for a fixed budget, allocate capacity to increasing the *number of agents* to diversify hypotheses, and keep $K$ modest (three to four at most) so that the central synthesis can reliably convert collective coverage into higher identification.

## D    Additional Experiments

### D.1    Phase-1 Diversity of Candidate Slates

**Experimental Setup.**    We analyze the diversity of Phase-1 candidate slates produced by different multi-agent collaboration frameworks on two reasoning benchmarks, AMC and GSM8K. For each problem and each method, we collect all final answers generated by the execution agents in the last collaboration round and compute three metrics: (i) **coverage rate**, the fraction of problems whose slate contains at least one correct final answer; (ii) **average distinct answers**, the average number of unique final answers in the slate after deduplication; and (iii) **inconsistency rate**, the fraction of problems for which different agents produce at least two distinct final answers, indicating explicit cross-agent disagreement. We compare MAESTRO against strong multi-agent baselines, including LLM-Debate, Dylan, and GPTSwarm, under the same backbone model and decoding configuration. In MAESTRO, each execution agent performs $K$-shot sampling in Phase 1, which naturally enlarges the candidate pool presented to the central selector.

**Results and Analysis.**    Table 5 summarizes the diversity statistics on AMC and GSM8K. On both datasets, MAESTRO attains the highest coverage rate (for example, 0.2892 vs. 0.1928 on AMC and 0.9325 vs. 0.8522 on GSM8K compared with the best baseline), which indicates that Phase-1 exploration is more effective at surfacing at least one correct solution in the slate. At the same time, MAESTRO produces substantially more distinct final answers per problem, with 7.55 vs. 2.86 on AMC and 2.92 vs. 2.05 on GSM8K relative to LLM-Debate, reflecting the impact of per-agent $K$-shot sampling. The inconsistency rates for MAESTRO are also high (0.98 on AMC and 0.80 on GSM8K), showing that agents frequently disagree and explore different hypotheses instead of collapsing to a single shared answer. Among the baselines, LLM-Debate tends to generate more diverse slates than Dylan and GPTSwarm, but still lags behind MAESTRO in both coverage and distinct answer counts. Overall, these results support that MAESTRO's Phase-1 module yields a rich, non-degenerate candidate pool with high coverage and substantial answer-level diversity, providing a strong starting point for the central selector in Phase 2.

### D.2    Controlling Phase-1 Diversity in MAESTRO

**Experimental Setup.**    We next study how MAESTRO's Phase-1 diversity can be explicitly controlled through simple knobs in the framework. We consider two families of ablations: (i) varying the number of execution agents $N$ while keeping the per-agent sampling depth $K$ fixed, and (ii) varying the per-agent sampling depth $K$ while keeping $N$ fixed.

**Results and Analysis.**    Tables 6 and 7 summarize the effect of agent count and sampling depth on MAESTRO's Phase-1 diversity. Varying the number of agents $N$ shows a clear trend on both datasets: more agents lead to higher coverage and more distinct answers in the slate. On AMC, coverage and distinct answers grow steadily as $N$ increases from 2 to 5, and GSM8K exhibits the same pattern. At the same time, inconsistency rates remain high, indicating persistent cross-agent disagreement rather than convergence to a single shared hypothesis. This suggests that additional agents contribute genuinely different candidate solutions instead of producing redundant copies.

Table 5: Phase-1 diversity metrics across multi-agent methods on AMC and GSM8K. *Coverage rate* is the fraction of problems whose candidate slate contains at least one correct answer. *Avg. distinct answers* is the average number of unique final answers per slate. *Inconsistency rate* is the fraction of problems where agents produce at least two different final answers.

| Dataset | Method | Coverage rate | Avg. distinct answers | Inconsistency rate |
|---------|--------|---------------|----------------------|--------------------|
| AMC | LLM-Debate | 0.1928 (±0.03) | 2.86 (±0.2) | 0.96 (±0.01) |
| | Dylan | 0.1807 (±0.01) | 2.77 (±0.3) | 0.93 (±0.02) |
| | GPTSwarm | 0.1325 (±0.02) | 2.48 (±0.1) | 0.83 (±0.01) |
| | MAESTRO | **0.2892** (±0.02) | 7.55 (±0.2) | 0.98 (±0.01) |
| GSM8K | LLM-Debate | 0.8522 (±0.01) | 2.05 (±0.4) | 0.68 (±0.04) |
| | Dylan | 0.8264 (±0.02) | 1.41 (±0.3) | 0.31 (±0.02) |
| | GPTSwarm | 0.8385 (±0.02) | 1.64 (±0.2) | 0.46 (±0.03) |
| | MAESTRO | **0.9325** (±0.01) | 2.92 (±0.4) | 0.80 (±0.03) |

Table 6: Effect of the number of agents on Phase-1 diversity metrics for MAESTRO on AMC and GSM8K.

| Dataset | Agents | Coverage rate | Avg. distinct answers | Inconsistency rate |
|---------|--------|---------------|----------------------|--------------------|
| AMC | 2 | 0.2678 (±0.02) | 5.20 (±0.5) | 0.98 (±0.01) |
| | 3 | 0.2892 (±0.03) | 7.55 (±0.3) | 0.98 (±0.01) |
| | 4 | 0.3614 (±0.01) | 9.69 (±0.2) | 0.99 (±0.01) |
| | 5 | **0.3735** (±0.01) | 11.98 (±0.2) | 0.94 (±0.02) |
| GSM8K | 2 | 0.9083 (±0.02) | 2.84 (±0.2) | 0.75 (±0.03) |
| | 3 | 0.9325 (±0.01) | 2.92 (±0.2) | 0.80 (±0.03) |
| | 4 | 0.9386 (±0.01) | 3.33 (±0.3) | 0.82 (±0.02) |
| | 5 | **0.9409** (±0.01) | 3.97 (±0.4) | 0.87 (±0.02) |

Changing the per-agent sampling depth $K$ yields a similar behaviour. Larger $K$ increases coverage and the average number of distinct answers on both AMC and GSM8K, with inconsistency rates staying in a high regime. In other words, drawing more samples per agent expands the candidate space and encourages more diverse final answers, while still maintaining a high probability that at least one correct solution is present in the slate. Together, these ablations show that MAESTRO exposes intuitive and effective controls for trading additional compute for higher coverage and diversity: widening the ensemble (larger $N$) and deepening stochastic exploration (larger $K$) both enlarge the candidate pool in a systematic way. The gains on GSM8K exhibit mild saturation at the highest $N$ and $K$, which is consistent with a regime where coverage is already very high and further improvements become harder.

### D.3 SENSITIVITY ANALYSIS OF $\lambda_{\text{RANK}}$

**Experimental Setup.** We evaluate the sensitivity of MAESTRO+CLPO to the rationale ranking weight $\lambda_{\text{rank}}$ on three benchmarks: GSM8K, AMC, and MMLU. The coefficient $\lambda_{\text{rank}}$ scales the ranking loss within the CLPO objective. When $\lambda_{\text{rank}} = 0$, the objective degenerates to a GRPO-style formulation without explicit ranking supervision. For each dataset, we train models with $\lambda_{\text{rank}} \in \{0, 0.1, 0.3, 0.5, 1.0\}$ under the same backbone, optimization hyperparameters, and MAESTRO collaboration settings, and we report end-to-end accuracy on the corresponding evaluation set. This experiment is designed to test whether CLPO is sensitive to the exact choice of $\lambda_{\text{rank}}$, and how much of the gain comes from moderate ranking weights compared to extreme values.

**Results and Analysis.** Table 8 shows that MAESTRO+CLPO is broadly robust to the choice of $\lambda_{\text{rank}}$ across all three benchmarks. On GSM8K and AMC, moving from $\lambda_{\text{rank}} = 0$ to a moderate value improves accuracy over the GRPO-style baseline, with both datasets peaking at $\lambda_{\text{rank}} = 0.5$ (from 0.8848 to 0.9030 on GSM8K and from 0.2530 to 0.3133 on AMC). Larger weights such as $\lambda_{\text{rank}} = 1.0$ lead to a small drop relative to the peak but still remain close to, or above, the $\lambda_{\text{rank}} = 0$

Table 7: Effect of per-agent sampling depth $K$ on Phase-1 diversity metrics for MAESTRO on AMC and GSM8K.

| Dataset | $K$ | Coverage rate | Avg. distinct answers | Inconsistency rate |
|---------|-----|---------------|----------------------|--------------------|
| AMC | 2 | 0.2651 | 5.10 | 0.94 |
| | 3 | 0.3494 | 7.55 | 0.98 |
| | 4 | 0.3735 | 9.78 | 0.98 |
| | 5 | **0.3855** | 12.47 | 0.99 |
| GSM8K | 2 | 0.9022 | 2.24 | 0.64 |
| | 3 | 0.9325 | 2.92 | 0.80 |
| | 4 | 0.9378 | 3.38 | 0.80 |
| | 5 | **0.9416** | 3.85 | 0.87 |

Table 8: Accuracy of MAESTRO+CLPO on GSM8K, AMC, and MMLU under different values of the rationale ranking weight $\lambda_{\text{rank}}$.

| Dataset | 0.0 | 0.1 | 0.3 | 0.5 | 1.0 |
|---------|-----|-----|-----|-----|-----|
| AMC | 0.2530 | 0.2771 | 0.2892 | **0.3133** | 0.2852 |
| GSM8K | 0.8848 | 0.8893 | 0.8954 | **0.9030** | 0.8933 |
| MMLU | 0.7132 | 0.7169 | 0.7201 | 0.7215 | **0.7238** |

setting, indicating that the method does not become unstable even when ranking supervision is strong. On MMLU, accuracy increases smoothly as $\lambda_{\text{rank}}$ grows, from 0.7132 at $\lambda_{\text{rank}} = 0$ to 0.7238 at $\lambda_{\text{rank}} = 1.0$, with only minor variation between intermediate values. Overall, these results suggest that CLPO does not rely on finely tuned values of $\lambda_{\text{rank}}$. A simple choice in a moderate range, for example between $0.3$ and $0.5$, is enough to obtain consistent gains over GRPO without demanding extensive hyperparameter search.

### D.4 LARGE LLM BACKBONE

**Experimental Setup.** To assess whether our collaboration framework and RL objectives transfer beyond small and mid-scale backbones, we evaluate MAESTRO on a larger model, LLAMA3-70B, on three representative benchmarks: GSM8K, AMC, and MMLU. For each dataset, we include a single-agent *Vanilla* prompting baseline, chain-of-thought prompting (CoT), self-consistency (SC), and several multi-agent collaboration baselines (LLM-Debate, GPTSwarm, and DYLAN). We then instantiate MAESTRO in its prompt-only form (central generation and selection) and in two RL-tuned variants, MAESTRO+GRPO and MAESTRO+CLPO, using the same collaboration structure (number of agents and rounds) as in the main experiments. All methods share the same LLAMA3-70B backbone, decoding configuration, and evaluation protocol, and we report exact match accuracy on each benchmark.

**Results and Analysis.** Table 9 shows that MAESTRO and its RL-tuned variants continue to provide consistent gains even when the backbone is as strong as LLAMA-3-70B. Moving from the single-agent *Vanilla* baseline to multi-agent methods already yields noticeable improvements. On GSM8K, GPTSwarm improves accuracy from 91.28 to 93.18. On AMC, LLM-Debate improves from 45.78 to 57.83. On MMLU, GPTSwarm reaches 78.35 from a vanilla score of 74.87. MAESTRO's prompt-only version further improves over these multi-agent baselines on all three datasets, achieving 93.86 on GSM8K, 59.04 on AMC, and 79.06 on MMLU.

RL tuning brings an additional and consistent boost. MAESTRO+GRPO increases GSM8K accuracy from 93.86 to 94.16 and AMC from 59.04 to 61.44, with a smaller but steady gain on MMLU. MAESTRO+CLPO achieves the best performance across all three benchmarks, reaching 94.85 on GSM8K, 63.86 on AMC, and 80.64 on MMLU. Relative to the vanilla 70B model, this corresponds to gains of 3.57, 18.08, and 5.77 points on GSM8K, AMC, and MMLU respectively, while also outperforming the strongest multi-agent baselines, for example by 1.67 points over GPTSwarm on GSM8K and 9.64 points over GPTSwarm on AMC. These results indicate that MAESTRO's collab-

Table 9: Performance (%) of LLAMA-3-70B on GSM8K, AMC, and MMLU under different prompting and multi-agent collaboration strategies. MAESTRO denotes the prompt-only central generation and selection framework, while MAESTRO+GRPO and MAESTRO+CLPO apply RL tuning on top of the same collaboration structure.

| Method | GSM8K | AMC | MMLU |
|---|---|---|---|
| Vanilla | 91.28 | 45.78 | 74.87 |
| CoT | 91.81 | 49.40 | 75.72 |
| SC | 92.57 | 54.21 | 77.28 |
| LLM-Debate | 92.87 | 57.83 | 78.08 |
| GPTSwarm | 93.18 | 54.22 | 78.35 |
| DYLAN | 92.27 | 56.63 | 77.18 |
| MAESTRO | 93.86 | 59.04 | 79.06 |
| MAESTRO + GRPO | 94.16 | 61.44 | 79.73 |
| **MAESTRO + CLPO** | **94.85** | **63.86** | **80.64** |

oration pattern and CLPO's rationale-aware objective remain effective at larger scales and continue to extract non-trivial improvements even in the high-accuracy regime of a strong foundation model.

### D.5 COMPUTE BUDGET AND TOKEN USAGE

**Experimental Setup.** We quantify the computational cost of MAESTRO and baseline methods by measuring token usage on GSM8K and AMC. For each method and dataset, we compute the average number of input tokens, output tokens, and their sum per problem over the full evaluation set. This gives a per-problem token budget that does not depend on the total number of evaluated problems. We then introduce an early-stopping strategy (for frameworks without a central agent, stopping once the majority vote over agents' answers converges across rounds; for frameworks with a central agent, stopping once the central selector's chosen candidate stabilizes across consecutive rounds) for all collaborative methods and re-run them with the same backbone and decoding configuration. For these runs we report both the solve rate (exact match accuracy) and the new average total tokens per problem. This setup lets us examine how expensive MAESTRO is under the full collaboration budget and how much of its accuracy gain can be preserved when we actively limit interaction depth.

**Results and Analysis.** Table 10 shows that MAESTRO has the highest average token usage when all planned rounds and samples are used. This is expected, since MAESTRO runs multiple agents, each drawing $K$ candidate solutions per round, and forwards all rationales to the central selector. On GSM8K, MAESTRO uses on average 12,458 tokens per problem, compared to 7,627 for GPTSwarm and 8,758 for LLM-Debate. On AMC, the pattern is similar. MAESTRO spends 24,282 tokens per problem, while LLM-Debate and GPTSwarm use 17,827 and 12,952 tokens respectively. These numbers reflect a deliberate design choice. MAESTRO allocates extra compute to Phase 1 exploration and centralized selection in order to improve coverage and identification, and this naturally raises the raw token budget.

Table 11 shows that early-stopping can substantially reduce MAESTRO's token cost while keeping most of its performance advantage. With early stopping, MAESTRO reaches a solve rate of 0.8789 on GSM8K with 8,172 tokens per problem on average. GPTSwarm attains 0.8372 with 6,385 tokens, and LLM-Debate reaches 0.8235 with 9,847 tokens. On AMC, MAESTRO achieves 0.2661 with 12,395 tokens, compared to 0.1968 with 11,525 tokens for Dylan and 0.1738 with 14,473 tokens for LLM-Debate. In both datasets, MAESTRO remains the most accurate multi-agent method, yet its token usage is now in the same order of magnitude as other strong baselines rather than being a clear outlier. Taken together, the two tables illustrate a flexible trade-off. The full-budget configuration shows the upper bound when compute is abundant, while the early-stopping configuration confirms that MAESTRO can operate in a more compute-conscious regime and still retain a clear accuracy margin over alternative collaboration schemes.

Table 10: Average token usage per problem. For each method we report the average input tokens, average output tokens, and their sum as the average total tokens per problem.

| Dataset | Method | Avg. input tokens | Avg. output tokens | Avg. total tokens |
|---------|--------|-------------------|--------------------|--------------------|
| GSM8K | Vanilla | 114 | 182 | 296 |
| | SC | 119 | 1,824 | 1,943 |
| | LLM-Debate | 6,974 | 1,784 | 8,758 |
| | Dylan | 4,135 | 3,476 | 7,611 |
| | GPTSwarm | 5,525 | 2,102 | 7,627 |
| | MAESTRO | 7,155 | 5,303 | 12,458 |
| AMC | Vanilla | 145 | 449 | 593 |
| | SC | 150 | 3,794 | 3,944 |
| | LLM-Debate | 13,968 | 3,859 | 17,827 |
| | Dylan | 8,208 | 3,317 | 11,525 |
| | GPTSwarm | 9,291 | 3,660 | 12,952 |
| | MAESTRO | 14,221 | 10,061 | 24,282 |

Table 11: Solve rate and average total tokens per problem with early-stopping strategy.

| Method | GSM8K | | AMC | |
|--------|-------|--|-----|--|
| | Solve rate | Avg. total tokens | Solve rate | Avg. total tokens |
| Vanilla | 0.7276 | 296 | 0.0803 | 593 |
| SC | 0.8079 | 1,943 | 0.1165 | 3,944 |
| LLM-Debate | 0.8235 | 9,847 | 0.1738 | 14,473 |
| Dylan | 0.8203 | 7,611 | 0.1968 | 11,525 |
| GPTSwarm | 0.8372 | 6,385 | 0.1284 | 9,362 |
| MAESTRO | **0.8789** | 8,172 | **0.2661** | 12,395 |

## D.6 WALL-CLOCK EFFICIENCY AND COMPUTE–PERFORMANCE TRADE-OFFS

**Experimental setup.** To quantify the compute–performance trade-off of MAESTRO relative to existing multi-agent frameworks, we measure wall-clock efficiency on GSM8K under a unified evaluation protocol. All methods run on the same hardware, with identical batch size and decoding configuration, and share the same Llama3-8B backbone. For each method, we time the end-to-end evaluation over the full GSM8K test set and report the average seconds per problem as well as the corresponding throughput (problems processed per second). We include single-agent baselines (Vanilla, self-consistency), prior multi-agent frameworks (Dylan, LLM-Debate, GPTSwarm), and three MAESTRO configurations that vary only in the number of agents (2, 3, or 4) while keeping the sampling depth and number of rounds fixed. This allows us to compare methods at matched wall-clock budgets and to trace MAESTRO's internal compute–performance curve as we scale collaboration.

**Results and analysis.** Table 12 and Figure 7 show that MAESTRO achieves favorable accuracy–efficiency trade-offs compared to both single-agent and multi-agent baselines. As expected, the single-agent Vanilla model is the fastest (0.26 s/problem, 3.79 problems/s) but also has the lowest solve rate (0.73). Adding self-consistency improves accuracy to 0.81 at roughly a 3× increase in wall-clock cost (0.72 s/problem). Multi-agent baselines further increase compute: Dylan and GPTSwarm require around 3.9 s/problem, yet their solve rates remain in the 0.82–0.85 range, while LLM-Debate sits in between at 2.05 s/problem and 0.8352 solve rate. In contrast, MAESTRO with two agents already attains a solve rate of 0.8693 at 2.57 s/problem, comparable in cost to LLM-Debate but with a clear accuracy gain. As we increase the number of agents, MAESTRO traces a smooth compute–performance frontier: the 3-agent configuration reaches 0.8933 solve rate at 4.26 s/problem, and the 4-agent configuration yields the highest accuracy of 0.9037 at 5.82 s/problem. Thus, for comparable or moderately higher wall-clock budgets than prior multi-agent methods, MAESTRO consistently delivers higher solve rates, and by scaling collaboration we obtain a

Table 12: Wall-clock efficiency and solve rate on GSM8K. We report the average seconds per problem, the corresponding throughput (problems per second), and the solve rate for all methods, measured on the *same* hardware and batch size.

| Method | Sec / problem | Problems / sec | Solve rate |
|---|---|---|---|
| Vanilla | 0.26 | 3.79 | 0.7276 |
| Self-Consistency (SC) | 0.72 | 1.40 | 0.8079 |
| Dylan | 3.91 | 0.26 | 0.8203 |
| LLM-Debate | 2.05 | 0.49 | 0.8352 |
| GPTSwarm | 3.89 | 0.26 | 0.8489 |
| MAESTRO (2 agents) | 2.57 | 0.39 | 0.8693 |
| MAESTRO (3 agents) | 4.26 | 0.24 | 0.8933 |
| MAESTRO (4 agents) | 5.82 | 0.17 | 0.9037 |

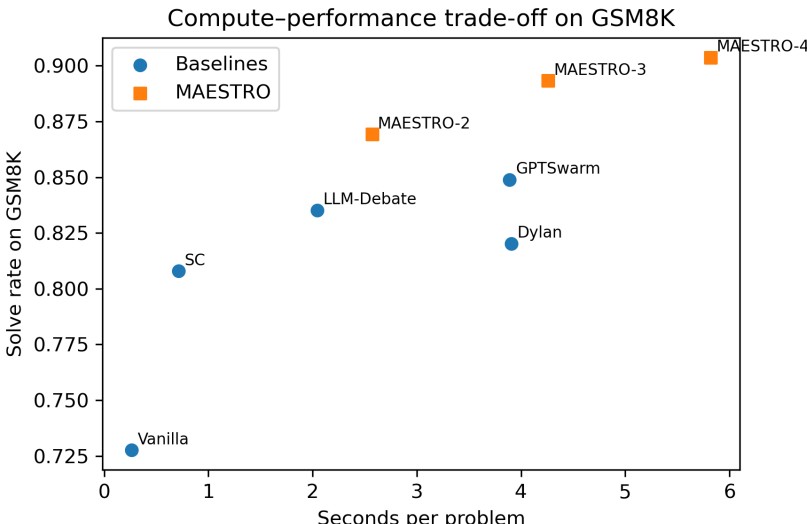

Figure 7: Compute–performance trade-off on GSM8K. The x-axis reports average seconds per problem (wall-clock cost), and the y-axis reports the solve rate. Blue circles denote single-agent and prior multi-agent baselines; orange squares denote different MAESTRO configurations with 2, 3, and 4 agents. MAESTRO variants trace a clear frontier that achieves higher accuracy than competing methods under comparable or moderately higher compute budgets.

controllable trade-off between additional compute and incremental performance gains, rather than relying on unbounded sampling."'

### D.7  Scaling with Agents and Rounds

**Experimental Setup.**   We examine how different collaboration frameworks behave when we increase the width and depth of interaction. On GSM8K and AMC, we first vary the number of execution agents $N \in \{2, 3, 4, 5, 6, 7\}$ while keeping the number of collaboration rounds fixed. We then vary the number of rounds $R \in \{2, 3, 4, 5\}$ under a fixed agent count. For each configuration we evaluate four representative multi-agent methods, LLM-Debate, DYLAN, GPTSwarm, and MAESTRO, using the same backbone model, decoding configuration, and evaluation protocol. We report the solve rate (exact match accuracy) on each benchmark. This setup is intended to reveal whether more agents and more rounds consistently help, or whether performance enters a regime of saturation and mild decline once collaboration becomes too wide or too deep.

Table 13: Impact of the number of agents on the solve rate of different multi-agent methods.

| Dataset | Method | 2 agents | 3 agents | 4 agents | 5 agents | 6 agents | 7 agents |
|---------|--------|----------|----------|----------|----------|----------|----------|
| GSM8K | LLM-Debate | 0.8241 | 0.8352 | 0.8309 | 0.8393 | 0.8286 | 0.8347 |
| | DYLAN | 0.8165 | 0.8203 | 0.8582 | 0.8521 | 0.8552 | 0.8628 |
| | GPTSwarm | 0.8067 | 0.8489 | 0.8628 | 0.8575 | 0.8514 | 0.8545 |
| | MAESTRO | **0.8693** | **0.8933** | **0.9037** | **0.9032** | **0.8961** | **0.8992** |
| AMC | LLM-Debate | 0.1807 | 0.1928 | 0.2289 | 0.2169 | 0.2048 | 0.2651 |
| | DYLAN | 0.1687 | 0.1968 | 0.2169 | 0.2410 | 0.2651 | 0.2538 |
| | GPTSwarm | 0.1566 | 0.1566 | 0.1446 | 0.1928 | 0.1566 | 0.1687 |
| | MAESTRO | **0.2530** | **0.2852** | **0.3052** | **0.2972** | **0.3133** | **0.3012** |

Table 14: Impact of the number of collaboration rounds on the solve rate of different multi-agent methods.

| Dataset | Method | 2 rounds | 3 rounds | 4 rounds | 5 rounds |
|---------|--------|----------|----------|----------|----------|
| GSM8K | LLM-Debate | 0.8241 | 0.8352 | 0.8287 | 0.8256 |
| | DYLAN | 0.8044 | 0.8203 | 0.8378 | 0.8453 |
| | GPTSwarm | 0.8234 | 0.8489 | 0.8036 | 0.7869 |
| | MAESTRO | **0.8789** | **0.8933** | **0.8872** | **0.8842** |
| AMC | LLM-Debate | 0.1687 | 0.1928 | 0.2410 | 0.2048 |
| | DYLAN | 0.1566 | 0.1968 | 0.2289 | 0.2169 |
| | GPTSwarm | 0.1325 | 0.1566 | 0.2048 | 0.2289 |
| | MAESTRO | **0.2851** | **0.2952** | **0.2892** | **0.2931** |

**Results and Analysis.** Tables 13 and 14 show that accuracy as a function of agent count and round count is non-monotonic for all methods rather than growing without bound. When increasing the number of agents, performance usually improves from two to three or four agents, then moves into a high plateau with small fluctuations. On GSM8K, GPTSwarm increases from 0.8067 (2 agents) to 0.8628 (4 agents), and MAESTRO rises from 0.8693 to 0.9037 over the same range. Beyond four agents, the gains saturate and scores oscillate in a narrow band. AMC shows a similar pattern, with MAESTRO improving from 0.2530 (2 agents) to 0.3052 (4 agents), then moving around this level for five to seven agents. Varying the number of rounds yields the same qualitative behaviour. Moving from two to three rounds often brings a clear gain, while additional rounds provide only small changes and can sometimes reduce accuracy, as seen for GPTSwarm on GSM8K when moving from three to five rounds. MAESTRO benefits from moderate values of $N$ and $R$ yet remains comparatively stable as collaboration becomes wider or deeper.

We believe this behaviour reflects a common coverage–noise trade-off in multi-agent reasoning. On math and symbolic reasoning tasks, a moderate number of agents and rounds is enough to reach high coverage and a strong selector. Once this regime is reached, extra agents mostly contribute redundant or noisy rationales, and longer interaction chains increase the chance that later messages introduce confusion rather than new useful evidence, especially under longer contexts. From the optimization side, our RL objective is designed to maximize end-to-end solve rate for a given collaboration pattern. It does not enforce that each additional agent or each extra round must improve performance. As a result, the empirical curves in Tables 13 and 14 are consistent with a picture where early increases in width and depth are valuable, and later increases mostly change the balance between additional signal and additional noise, leading to the observed plateau with mild, non-systematic variations rather than a sharp scalability limit.

### D.8 COVERAGE–IDENTIFICATION DYNAMICS

**Experimental Setup.** We study how MAESTRO's coverage and identification behaviour changes with the depth of collaboration and with CLPO training. On GSM8K and AMC, we fix the number of agents and the per-agent sampling depth, and vary the maximum number of rounds $R \in \{2, 3, 4, 5\}$.

Table 15: Coverage probability $p_t$, identification probability $q_t$, and accuracy Acc at the final collaboration round $t = R$ for different numbers of rounds on GSM8K and AMC.

| Dataset | Rounds | $p_t$ | $q_t$ | Acc |
|---------|--------|-------|-------|-----|
| GSM8K | 2 | 0.9272 | 0.9479 | 0.8789 |
| | 3 | 0.9325 | **0.9580** | **0.8933** |
| | 4 | **0.9417** | 0.9422 | 0.8872 |
| | 5 | 0.9280 | 0.9528 | 0.8842 |
| AMC | 2 | 0.3395 | 0.8397 | 0.2851 |
| | 3 | 0.3494 | 0.8449 | **0.2952** |
| | 4 | 0.3412 | **0.8477** | 0.2892 |
| | 5 | **0.3503** | 0.8368 | 0.2931 |

For each $R$, we run the full MAESTRO protocol and focus on the final collaboration round $t = R$. At this round, we estimate the empirical coverage probability $p_t$, defined as the fraction of problems whose Phase 1 slate contains at least one correct candidate solution, and the empirical identification probability $q_t$, defined as the fraction of covered problems where the central policy selects a correct candidate from the slate. We also report the final solve rate Acc. In a second study on GSM8K, we fix the collaboration pattern and track $(p_t, q_t, \text{Acc})$ at several CLPO training epochs. This design instantiates the coverage–identification view both across collaboration depth and across training time.

**Results and Analysis.** Table 15 reports $p_t$, $q_t$, and Acc at the final round for different numbers of rounds. On GSM8K, moving from two to three rounds improves both identification and final performance. Coverage increases slightly from $p_2 = 0.9272$ to $p_3 = 0.9325$, identification rises from $q_2 = 0.9479$ to $q_3 = 0.9580$, and accuracy improves from $0.8789$ to $0.8933$. Beyond three rounds, coverage remains high and even peaks at $p_4 = 0.9417$, but identification no longer improves in a monotone way and accuracy shows a small decline to $0.8872$ at four rounds and $0.8842$ at five rounds. On AMC, increasing the number of rounds from two to three yields the best trade-off, with coverage rising from $p_2 = 0.3395$ to $p_3 = 0.3494$, identification from $q_2 = 0.8397$ to $q_3 = 0.8449$, and accuracy from $0.2851$ to $0.2952$. At four and five rounds, coverage and identification move in opposite directions and accuracy fluctuates in a narrow band around the three-round value. Together, these trends match the coverage–noise trade-off suggested by our theory. Early rounds help agents share information, lift coverage, and sharpen the selector. Once $p_t$ and $q_t$ are already high, additional rounds mainly introduce redundant or noisy rationales, so gains saturate and small oscillations in $p_t$ and $q_t$ translate into marginal changes in Acc. This supports the choice of using a small number of rounds, such as three, which captures most of the attainable improvement without unnecessary collaboration overhead.

Table 16 shows how $(p_t, q_t, \text{Acc})$ evolve across CLPO training epochs on GSM8K for a fixed collaboration pattern. Coverage starts high and remains in a narrow band between $0.93$ and $0.94$, indicating that Phase 1 already surfaces a correct candidate for most problems before CLPO is applied. In contrast, identification improves steadily from $q_t = 0.9253$ at epoch 0 to $q_t = 0.9580$ at epoch 3, and accuracy follows the same trend, moving from $0.8732$ to $0.8933$. This behaviour matches the intended role of CLPO, which is to refine the central selector rather than to radically change exploration. It also provides a concrete link between the coverage–identification inequality and practice: training mainly increases $q_t$ while keeping $p_t$ high, which raises the effective $p_t q_t$ term and leads to higher solve rates, with diminishing gains once both quantities enter a high regime.

## D.9 COMPARISON WITH RL-TUNED MULTI-AGENT BASELINES

**Experimental Setup.** We compare MAESTRO with two recent RL-tuned multi-agent frameworks that explicitly train agents to coordinate. MAS-GPT formulates multi-agent system construction as a generative language problem. Given a user query, the model produces an executable MAS specification in a single forward pass, which is then executed to solve the task. ReMA adopts a hierarchical design with a high-level meta-thinking agent that plans and monitors and a low-level reasoning agent that carries out detailed inference. Both methods rely on reinforcement learning rather than pure

Table 16: Evolution of coverage probability $p_t$, identification probability $q_t$, and accuracy Acc across CLPO training epochs on GSM8K.

| Epoch | $p_t$ | $q_t$ | Acc |
|---|---|---|---|
| 0 | 0.9437 | 0.9253 | 0.8732 |
| 1 | 0.9365 | 0.9468 | 0.8867 |
| 2 | 0.9271 | 0.9605 | 0.8905 |
| 3 | 0.9325 | 0.9580 | 0.8933 |

Table 17: Comparison with RL-tuned multi-agent baselines on GSM8K, AMC, MMLU, and HumanEval. All methods use the same LLAMA-3-8B backbone and evaluation protocol.

| Method | GSM8K | AMC | MMLU | HumanEval |
|---|---|---|---|---|
| ReMA | 0.8719 | 0.2048 | 0.7125 | 0.6463 |
| MAS-GPT | 0.8635 | 0.2771 | 0.7096 | **0.6829** |
| MAESTRO | **0.8933** | **0.2852** | **0.7238** | 0.6687 |

prompting to shape collaboration. We evaluate MAS-GPT, ReMA, and MAESTRO on four benchmarks: GSM8K (math word problems), AMC (competition-style math), MMLU (broad knowledge and reasoning), and HumanEval (code generation). All methods use the same LLAMA-3-8B backbone, decoding configuration, and evaluation protocol, and we report exact match accuracy on GSM8K, AMC, and MMLU and pass@1 on HumanEval.

**Results and Analysis.** Table 17 summarizes the comparison. MAESTRO attains the strongest performance on three of the four benchmarks. On GSM8K it reaches 0.8933 accuracy, improving over ReMA (0.8719) and MAS-GPT (0.8635). On AMC, which is more challenging and less saturated, MAESTRO achieves 0.2852 and thus exceeds MAS-GPT (0.2771) and substantially surpasses ReMA (0.2048). On MMLU, MAESTRO again leads with 0.7238, compared to 0.7125 for ReMA and 0.7096 for MAS-GPT, indicating that its collaboration pattern and RL objective transfer to broad knowledge benchmarks rather than overfitting to math-focused tasks. On HumanEval, MAS-GPT attains the highest score (0.6829), which is consistent with its design as a code-centric MAS generator, while MAESTRO remains competitive at 0.6687 and ahead of ReMA (0.6463). Overall, MAESTRO is competitive with these specialized RL-based multi-agent frameworks and often stronger. This supports the view that coverage–identification guided collaboration offers a complementary way to structure coordination, alongside programmatic MAS generation in MAS-GPT and hierarchical meta-thinking in ReMA, and that it yields robust gains in multi-task settings under a shared backbone.

D.10    GENERALIZATION TO SCIENTIFIC QUESTION ANSWERING

**Experimental Setup.** To test whether MAESTRO and its RL-tuned variants generalize beyond math, coding, and broad knowledge benchmarks, we further evaluate on GPQA, a challenging multiple-choice scientific QA dataset. GPQA contains 448 expert-written questions spanning biology, physics, and chemistry. The questions are deliberately difficult and designed to resist simple lookup strategies. PhD-level experts achieve around 65% accuracy, while strong non-experts with unrestricted web access reach only about 34%. This setting is therefore closer to realistic "Google-proof" scientific oversight, where models must reason carefully rather than retrieve surface-level facts. We evaluate a single-agent *Pure* prompting baseline, chain-of-thought prompting (CoT), self-consistency (SC), and several multi-agent collaboration baselines (LLM-Debate, Dylan, GPTSwarm), together with our MAESTRO framework in prompt-only form and two RL-tuned variants, MAESTRO+GRPO and MAESTRO+CLPO. All methods use the same backbone model and decoding configuration, and we report accuracy on the GPQA multiple-choice questions.

**Results and Analysis.** Table 18 reports the results on GPQA. Multi-agent collaboration consistently improves over single-agent prompting. Moving from Pure (0.2677) to CoT (0.2828) and SC (0.3081) yields moderate gains, and adding explicit multi-agent debate further improves accuracy,

Table 18: Accuracy on the GPQA scientific QA benchmark for single-agent and multi-agent methods, including MAESTRO with and without RL tuning.

| Method | GPQA |
|---|---|
| Pure | 0.2677 |
| CoT | 0.2828 |
| SC | 0.3081 |
| LLM-Debate | 0.3383 |
| Dylan | 0.3232 |
| GPTSwarm | 0.3434 |
| MAESTRO | 0.3636 |
| MAESTRO + GRPO | 0.3788 |
| **MAESTRO + CLPO** | **0.3940** |

Table 19: Accuracy on GSM8K under different convergence strategies. The left block varies the number of execution agents using LLaMA3-8B. The right block varies the backbone with a fixed collaboration structure.

| Method | Vary #agents | | | Vary backbone | | |
|---|---|---|---|---|---|---|
| | 3 agents | 5 agents | 7 agents | LLaMA3-8B | LLaMA3-3B | Qwen-3B |
| Central-Gen | 0.8393 | 0.8638 | 0.8669 | 0.8393 | 0.7961 | 0.8832 |
| Central-Gen+SC | 0.8467 | 0.8853 | 0.8937 | 0.8467 | 0.8226 | 0.8916 |
| Central-Gen+Select | 0.8703 | 0.9032 | 0.8992 | 0.8703 | 0.8153 | 0.9083 |

with LLM-Debate reaching 0.3383 and GPTSwarm reaching 0.3434. MAESTRO's prompt-only variant attains 0.3636, outperforming all prompt-based and multi-agent baselines, which indicates that its coverage and centralized selection mechanism remain effective on hard scientific QA. RL tuning then provides a further boost. MAESTRO+GRPO reaches 0.3788, and MAESTRO+CLPO achieves the best performance at 0.3940. The margin over the strongest baseline GPTSwarm is about five percentage points, which is substantial in this high-difficulty regime where even non-expert humans struggle. These findings suggest that MAESTRO and CLPO do not overfit to math-style reasoning tasks but transfer to a distinct, expert-level QA domain where scalable oversight and robust identification of correct answers are especially important.

### D.11 EFFECT OF CANDIDATE POOL SIZE AND BACKBONE STRENGTH ON CENTRAL SELECTION

**Experimental Setup.** We examine how the advantage of the central selection mechanism behaves when the candidate pool grows larger and when the execution agents are powered by weaker backbones. All experiments in this section use the GSM8K benchmark. We consider three convergence variants: **Central-Gen**, where the central agent directly generates a final solution; **Central-Gen+SC**, where the central agent applies self-consistency over multiple generations; and **Central-Gen+Select**, where the central agent only scores and selects from Phase-1 candidates without regenerating new chains. In the first setting, we fix the backbone to LLaMA3-8B and vary the number of execution agents. In the second setting, we fix the collaboration structure and vary the backbone among LLaMA3-8B, LLaMA3-3B, and Qwen-3B.

**Results and Analysis.** Table 19 reveals two consistent trends. First, when we fix the backbone to LLaMA3-8B and increase the number of agents, all three convergence variants benefit from the larger candidate pool, but **Central-Gen+Select** remains the strongest across 3, 5, and 7 agents. As we add more agents and thus more (potentially noisy) candidates, both self-consistency and selection gain from the extra hypotheses; however, selection maintains a clear edge, which indicates that a ranking-based convergence stage can still reliably extract good answers even when the pool becomes larger and more redundant.

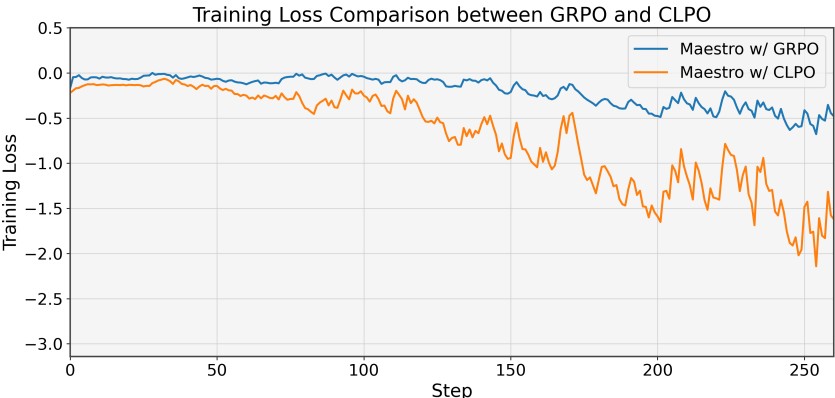

Figure 8: Training loss comparison between MAESTRO with GRPO and MAESTRO with CLPO during RL fine-tuning on GSM8K. Both runs share the same SFT initialization and hyperparameters, yet CLPO reaches a lower and more steadily decreasing objective, indicating more effective use of ranking-based feedback.

Second, when we fix the collaboration structure and vary the backbone, the advantage of selection becomes more pronounced as the central policy grows stronger. With LLaMA3-8B and Qwen-3B, Central-Gen+Select outperforms both Central-Gen and Central-Gen+SC, for example improving over self-consistency from 0.8467 to 0.8703 on LLaMA3-8B and from 0.8916 to 0.9083 on Qwen-3B. For the weaker LLaMA3-3B model, self-consistency is slightly better than selection (0.8226 vs. 0.8153), which matches the intuition that a weaker central policy can benefit from voting over several fresh generations. In contrast, a stronger central policy can more effectively rank and filter the Phase-1 candidates without regenerating new chains. Since self-consistency always requires sampling multiple new rationales in the convergence stage, it incurs a much higher token cost than selection. Taken together, these results suggest that once the central policy has moderate strength, central selection remains robust under noisier candidate pools and typically offers better or comparable accuracy at a substantially lower computational budget.

### D.12 TRAINING DYNAMICS OF GRPO VS. CLPO

**Experimental setup.** To examine the optimization behavior of our objective, we compare the training trajectories of MAESTRO when fine-tuned with GRPO and with CLPO. Both runs start from the same SFT-initialized backbone and use GSM8K as the training task. We keep all RL hyperparameters identical, including learning rate, batch size, KL regularization, and rollout configuration. The only difference is the choice of RL objective. During training, we record the scalar objective value at each update step and track its evolution over time.

**Results and analysis.** Figure 8 shows the resulting loss curves. Both GRPO and CLPO exhibit a short warm-up phase in which the objective fluctuates around zero. After this initial period, the GRPO curve quickly settles into a narrow band and plateaus, with the loss remaining close to a small negative value for the rest of training. In contrast, the CLPO curve continues to decrease in a smooth and stable way, reaching substantially lower objective values without signs of divergence or strong oscillation. Since the objective is defined up to a baseline and more negative values correspond to higher advantages assigned to good trajectories, this pattern indicates that CLPO extracts richer learning signal and drives the policy toward solutions that better separate high-quality from low-quality rationales. This behavior is consistent with our coverage–identification view: GRPO saturates early, whereas CLPO keeps improving the selector's identification probability $q_t$ and therefore continues to make meaningful progress throughout training.

# E  DECLARATION ON THE USE OF LARGE LANGUAGE MODELS

In the preparation of this work, the authors used GPT-5 and GPT-4o for two specific purposes. First, GPT-5 was employed to polish the writing, improve clarity, and ensure grammatical correctness throughout the manuscript. Second, GPT-4o was used during dataset construction to assist in evaluating the quality of reasoning annotations. After using these tools, the authors reviewed and edited all content as needed and take full responsibility for the final version of the publication.

