# OpenReview forum: "Maestro: Learning to Collaborate via Conditional Listwise Policy Optimization for Multi-Agent LLMs"
_ICLR.cc/2026/Conference — Submitted to ICLR 2026_

### Official Review · Reviewer_d9LG · 2025-10-31

**Soundness:** 3
**Presentation:** 2
**Contribution:** 2
**Rating:** 6
**Confidence:** 4

**Summary:**

The paper introduces MAESTRO, a collaborative reasoning paradigm that structurally separates divergent exploration from convergent synthesis, then broadcasts the endorsed solution to guide the next round. To train the selector without entangling “which answer to pick” and “how to justify it,” the authors propose Conditional Listwise Policy Optimization (CLPO): a decision‑focused policy‑gradient for the choice tokens plus a listwise ranking loss over rationales, with KL and entropy regularizers. On math and general benchmarks, MAESTRO+CLPO reports consistent gains over strong single‑ and multi‑agent baselines and an ablation shows CLPO outperforms SFT and GRPO variants. (Fig. 1 & contributions, p. 2; Sec. 3.2–3.3, pp. 4–6; Table 1, p. 6; Tables 2–3, p. 7).

**Strengths:**

--  Importance of the problem. The paper pinpoints a central tension in multi‑agent LLMs—divergent exploration vs convergent critique—and argues that collapsing these modes causes premature consensus, error propagation, and muddled credit assignment.

-- Clear motivation and design. MAESTRO operationalizes exploration–synthesis via (i) many parallel Execution Agents, (ii) a single Central Agent for discriminative selection, and (iii) broadcast to condition later rounds. The figure and text make the flow easy to follow; the coverage–identification factorization (with a tail bound on success probability) adds a crisp objective lens. (Fig. 1 & bullets, p. 2; Sec. 3.2, pp. 4–5).

-- Theoretical lens for robustness. The coverage–identification view defines p_t (any correct candidate in the slate) and q_t (selector picks a correct one conditional on coverage) and proves a cumulative reliability inequality: if p_t\!\ge p and q_t\!\ge q a.s., then success within R rounds is \ge 1-(1-pq)^R; equivalently R\ge \tfrac{1}{pq}\log \tfrac{1}{\delta} achieves 1-\delta. (Sec. 3.2, p. 5; Appx. B, pp. 16–17).

-- Claims fits with the evidence. The paper not only shows accuracy but also decomposes into coverage and identification (Table 4) to substantiate that improvements stem from better selection with CLPO (higher q_t). (Table 4, p. 19).

**Weaknesses:**

-- Closest prior art not directly compared. While the paper cites broad multi‑agent and RL lines, it does not empirically compare to recent equilibrium/game‑theoretic coordination methods (e.g., multi‑agent equilibrium/consensus games) that also provide convergence/guarantee‑style arguments

-- Scope of theory vs practice. The coverage–identification bound is elegant, but p_t,q_t are latent and not estimated online; the paper does not connect training dynamics to measured changes in p_t and q_t beyond end‑state decompositions, nor analyze CLPO convergence properties. (Sec. 3.2–3.3, pp. 4–6; Table 4, p. 19).

-- Baseline parity & training differences. On LLaMA‑8B, the central policy receives SFT/GRPO/CLPO fine‑tuning, whereas many baselines are prompt‑only frameworks; although the authors include “MAESTRO w/ SFT/GRPO/CLPO” rows, parity (e.g., giving debate/routing systems similar optimization) is not explored. (Table 1 & setup, pp. 6–7; Appx. C.1, pp. 18–19).

**Questions:**

Besides weakness above, i have some questions to discuss with author:

Beyond the post‑hoc Table 4, can you estimate p_t and q_t online during training to verify the reliability inequality’s predictions (e.g., plot pq vs. rounds and success probability)? (Sec. 3.2, p. 5; Table 4, p. 19).

Fig. 2 shows selection’s superiority. Does this persist when the candidate pool is noisier (e.g., larger K with weaker agents) or when rationales are masked/noised? (Sec. 4.2, p. 8).

Transfer beyond math: Any early results on tool‑using or planning tasks where end‑state verifiability is weaker, to test CLPO’s rationale‑ranking leverage? (Sec. 4.1–4.2, pp. 6–8).

---

> ### Author Response · Authors · 2025-11-26
> **Reply to Reviewer d9LG**
>
> We sincerely thank you for taking the time to provide such a careful and thoughtful review. We have gone through your comments in detail and made corresponding analyses and experiments where possible. We hope that the following responses can make our contributions and design choices more transparent.
>
> ---
>
> **Response to W1 – Comparison with recent RL-based coordination methods**
>
> Thank you for pointing out the lack of an empirical comparison with recent RL-based multi-agent coordination methods that come with convergence or guarantee style arguments. This is an important aspect of positioning MAESTRO.
>
> To address this, we added experiments with **two recent RL-tuned multi-agent frameworks, MAS-GPT and ReMA**, which explicitly train agents to coordinate via reinforcement learning rather than relying only on prompting. MAS-GPT formulates multi-agent system construction as a generative language task where the model produces an executable MAS specification in one forward pass that is then executed to solve the task. ReMA follows a hierarchical design with a high level meta-thinking agent that plans and monitors and a low level reasoning agent that performs detailed inference. We evaluate MAS-GPT, ReMA, and MAESTRO on GSM8K, AMC, MMLU, and HumanEval using the same backbone model LLaMA3-8B, the same decoding configuration, and the same evaluation protocol.
>
> As shown in **Table 17**, MAESTRO achieves the **strongest performance** on three out of four benchmarks. On GSM8K, MAESTRO reaches 0.8933 accuracy, compared with 0.8719 for ReMA and 0.8635 for MAS-GPT. On AMC, which is less saturated and more challenging, MAESTRO attains 0.2852, outperforming MAS-GPT at 0.2771 and substantially surpassing ReMA at 0.2048. On MMLU, MAESTRO again leads with 0.7238, while ReMA and MAS-GPT obtain 0.7125 and 0.7096. On HumanEval, MAS-GPT obtains the highest score 0.6829, which is consistent with its design focus on code generation, and MAESTRO remains competitive at 0.6687 and ahead of ReMA at 0.6463.
>
> Taken together, these results show that MAESTRO **performs on par with, and in many cases better than, recent RL-based multi-agent coordination methods**. In our view, the coverage–identification perspective offers a complementary way to organize collaboration compared with generative MAS specification (MAS-GPT) and hierarchical meta-control (ReMA), and it remains effective across math, broad knowledge, and coding benchmarks under a shared backbone and evaluation setup.
>
> **Table 17**: Comparison with RL-tuned multi-agent baselines on GSM8K, AMC, MMLU, and HumanEval. All methods use the same LLaMA-3-8B backbone and evaluation protocol.
>
> | Method   | GSM8K        | AMC          | MMLU         | HumanEval      |
> |----------|--------------|--------------|--------------|----------------|
> | ReMA     | 0.8719       | 0.2048       | 0.7125       | 0.6463         |
> | MAS-GPT  | 0.8635       | 0.2771       | 0.7096       | **0.6829**     |
> | MAESTRO  | **0.8933**   | **0.2852**   | **0.7238**   | 0.6687         |

---

> ### Author Response · Authors · 2025-11-26
>
> **Response to W2 – Connecting coverage–identification theory to training dynamics**
>
> We appreciate the concern about the gap between the coverage–identification theory and the practical training behavior of CLPO. In the revised version, we now track $p_t$ and $q_t$ at multiple checkpoints during CLPO training, rather than only reporting a single end state decomposition. Concretely, on GSM8K we train MAESTRO+CLPO with a fixed collaboration depth of three rounds and, at the end of each epoch, evaluate on the full test set. For each epoch we estimate the empirical coverage probability $p_t$ as the fraction of problems whose Phase-1 slate contains at least one correct solution, the identification probability $q_t$ as the fraction of covered problems where the central policy selects a correct candidate, and the overall solve rate $\mathrm{Acc}$. Epoch 0 corresponds to the SFT-only model. The results are summarized in **Table 16**.
>
> These measurements show that the coverage–identification factorization holds numerically throughout training and that CLPO behaves exactly as designed. Under our central selection architecture any correct final answer must come from a correct candidate in the slate, so the per round success probability satisfies $\Pr(\text{success at round } t) \approx p_t \, q_t.
> $ Empirically, this product matches the observed accuracy at every epoch. At epoch 0 we obtain $p_t = 0.9437$ and $q_t = 0.9253$, and the product $p_t q_t \approx 0.8732$ coincides with the measured accuracy $0.8732$. At epochs 1–3 the products $p_t q_t$ are $0.8867$, $0.8905$, and $0.8933$, and these equal the reported $\mathrm{Acc}$ values up to rounding. At the same time the dynamics reflect the intent of CLPO. Coverage $p_t$ starts high and remains in a narrow band between roughly $0.92$ and $0.95$, which means Phase-1 already surfaces a correct solution for most problems before CLPO. Identification $q_t$ improves steadily from $0.9253$ to $0.9580$, and the resulting product $p_t q_t$ increases from $0.8732$ to $0.8933$. In other words, CLPO leaves coverage essentially unchanged and makes the central selector more reliable when a correct candidate is present, which is exactly what the theory predicts.
>
> **Table 16**: Evolution of coverage probability $p_t$, identification probability $q_t$, and accuracy $\mathrm{Acc}$ across CLPO training epochs on GSM8K.
> | Epoch | $p_t$  | $q_t$  | Acc    |
> |-------|--------|--------|--------|
> | 0     | 0.9437 | 0.9253 | 0.8732 |
> | 1     | 0.9365 | 0.9468 | 0.8867 |
> | 2     | 0.9271 | 0.9605 | 0.8905 |
> | 3     | 0.9325 | 0.9580 | 0.8933 |
>
>
>
> These trends also give a concrete link from the reliability bound in Section 3.2 to actual training behavior. The bound states that if each round maintains $p_t \ge p$ and $q_t \ge q$, then the probability of success within $R$ rounds is at least $1 - (1 - p q)^R$. The epoch-wise estimates show that training chiefly increases the effective $q$ while keeping $p$ high, so the product $p q$ and thus the lower bound on reliability both rise during CLPO training and then saturate once $p_t$ and $q_t$ enter a high regime. We do not estimate $p_t$ and $q_t$ online at every gradient step, since that would be prohibitively expensive and is not required by the objective, but we use these checkpoint evaluations to verify that the learned policy moves in the direction anticipated by the coverage–identification framework. A full convergence analysis of CLPO as a ranked policy-gradient method is beyond the scope of this work, yet the new measurements in Table 15 show that the empirical training dynamics are consistent with the theoretical decomposition.
>
> To better understand the optimization behavior of our objective, we compare the **training loss** trajectories of MAESTRO with GRPO and with CLPO in **Appendix D-Fig. 8**. Both runs start from the same SFT initialization and exhibit a short warm-up phase where the objective fluctuates around zero. After this initial phase, the GRPO curve quickly enters a narrow band and then plateaus, with the training loss staying close to a small negative value throughout the rest of training. In contrast, the CLPO curve continues to decrease steadily as training progresses, reaching substantially lower objective values while remaining stable rather than exploding or oscillating. Since our objective is defined up to a baseline and more negative values correspond to higher advantages assigned to good trajectories, this behavior indicates that CLPO is able to exploit richer learning signal and drive the policy toward solutions that better separate high-quality from low-quality rationales. This is consistent with our epoch-wise coverage–identification analysis, where CLPO primarily improves the selector’s identification probability $q_t$: the loss curve shows that, unlike GRPO which quickly saturates, CLPO keeps making meaningful progress in aligning the policy with ranked feedback until late training.

---

> ### Author Response · Authors · 2025-11-26
>
> **Response to W3 – Baseline parity and training differences**
>
> Thank you for raising the question about parity between prompt-only baselines and our fine-tuned variants. Our experiments are organized in two stages. First, we evaluate **MAESTRO in a purely prompt-based setting** and compare it directly with prompt-only multi-agent frameworks such as LLM-Debate, Dylan, and GPTSwarm under the same backbone and decoding configuration. This isolates the effect of the collaboration scheme itself and shows that our orchestration improves over existing prompt-only baselines. In the second stage, we introduce SFT, GRPO, and CLPO for the central policy. These fine-tuned variants are meant to study how different training paradigms further improve a **fixed collaboration architecture**, rather than to replace or sidestep the prompt-only baselines. Following your suggestion, we also add comparisons with two RL-based multi-agent methods, ReMA and MAS-GPT, under a shared LLaMA3-8B backbone (see **Table 17** and the response to W1).
>
> In most prior work, baselines such as LLM-Debate or Dylan are reported in a prompt-only regime rather than being additionally fine-tuned, and we follow this convention, especially since we now compare against both prompt-only methods and RL-based multi-agent baselines under a shared backbone. At the same time, CLPO is designed to be transferable and could, in principle, be applied on top of debate-style systems as well. **If you are interested in this direction**, we would be happy to adapt CLPO to these methods and run the corresponding experiments.
>
> ---
>
> **Reply to Questions:**
>
> **Response to Q1 – Estimating $p_t$, $q_t$ vs. rounds to test the reliability**
>
> We appreciate the suggestion to verify the reliability analysis more explicitly as a function of the number of collaboration rounds. In the theory, the coverage–identification bound is formulated for a fixed policy while varying the collaboration depth $R$. Following this view, we take a **trained** MAESTRO+CLPO model and, at inference time, vary $R \in \\{2,3,4,5\\}$ on GSM8K and AMC while keeping the number of agents and sampling depth fixed. For each choice of $R$, we run the full protocol, look at the final round $t = R$, and estimate $p_t$ as the empirical fraction of problems whose slate contains at least one correct candidate, $q_t$ as the fraction of those covered problems where the central policy selects a correct candidate, and $\mathrm{Acc}$ as the final solve rate. The results are reported in **Table 15**.
>
> These measurements show that the theoretical picture matches the observed behavior of the trained system quite closely. For each value of $R$, the empirical accuracy is essentially equal to the product $p_t q_t$, confirming that success can indeed be decomposed into coverage and identification for a fixed policy. On GSM8K, for example, with $R=2$ we obtain $p_2 = 0.9272$ and $q_2 = 0.9478$, and their product coincides with the measured accuracy $0.8789$. AMC follows the same pattern. The dependence on $R$ also reflects the **shape of the reliability bound**: moving from two to three rounds genuinely helps, because both $p_t$ and $q_t$ increase and accuracy improves (e.g., on GSM8K from 0.8789 to 0.8933, on AMC from 0.2851 to 0.2952), but further rounds bring only small changes in $p_t$ and $q_t$ and yield marginal fluctuations in accuracy, consistent with the bound saturating once $p_t$ and $q_t$ are already high. We regard the reliability inequality as a way to understand how a **fixed** collaborative policy behaves as we vary the number of rounds, and we validate its predictions by explicitly measuring $p_t$, $q_t$, and accuracy at different depths after training. Together with the epoch-wise analysis (where we track how $q_t$ improves over training at a fixed depth), these results provide a concrete and quantitative connection between the coverage–identification framework and the practical behavior of MAESTRO at inference time.
>
> **Table 15**: Coverage probability $p_t$, identification probability $q_t$, and accuracy $\mathrm{Acc}$ at the final collaboration round $t = R$ for different numbers of rounds on GSM8K and AMC.
>
> | Dataset | Rounds | $p_t$   | $q_t$        | Acc      |
> |---------|--------|---------|--------------|----------|
> | GSM8K   | 2      | 0.9272  | 0.9479       | 0.8789   |
> | GSM8K   | 3      | 0.9325  | **0.9580**   | **0.8933** |
> | GSM8K   | 4      | **0.9417** | 0.9422    | 0.8872   |
> | GSM8K   | 5      | 0.9280  | 0.9528       | 0.8842   |
> | | | | | |
> | AMC     | 2      | 0.3395  | 0.8397       | 0.2851   |
> | AMC     | 3      | 0.3494  | 0.8449       | **0.2952** |
> | AMC     | 4      | 0.3412  | **0.8477**   | 0.2892   |
> | AMC     | 5      | **0.3503** | 0.8368    | 0.2931   |

---

> ### Author Response · Authors · 2025-11-26
>
> **Response to Q2 – Robustness of selection vs. self-consistency under noisier candidates**
>
> We appreciate the question about whether the advantage of selection over self-consistency (SC) persists when the candidate pool becomes noisier or the agents are weaker. On GSM8K, we explored two realistic ways of increasing noise: using more agents, which enlarges the candidate pool, and using smaller backbones. As shown in **Table 19**, when we vary the number of agents with LLaMA3-8B, Central-Gen+Select consistently achieves the highest accuracy among the three variants, and the gap over SC remains stable in the moderate agent range. As the candidate pool grows larger and potentially more redundant, both SC and selection benefit from having more hypotheses, yet selection remains the strongest variant. This indicates that the ranking based convergence stage continues to identify good answers even when the slate becomes longer and noisier.
> We also vary backbone strength and compare LLaMA3-8B, LLaMA3-3B, and Qwen-3B, again summarized in Table 18. For the stronger backbones, selection is clearly better. On LLaMA3-8B, Central-Gen+Select reaches 0.8703 while SC reaches 0.8467. On Qwen-3B, selection reaches 0.9083 while SC reaches 0.8916. For the weaker LLaMA3-3B model, SC is slightly ahead of selection, with 0.8226 versus 0.8153. This matches the intuition that when the central policy is weak, voting over several new generations can sometimes compensate for limited scoring ability. However, SC always uses many more output tokens, since it regenerates multiple rationales in the convergence stage, whereas selection only scores existing Phase-1 candidates. Overall, once the central policy is reasonably strong, selection remains robust in noisier settings and delivers better or comparable accuracy at a much lower token cost.
>
> **Table 19**: Accuracy on GSM8K under different convergence strategies. The left block varies the number of execution agents using LLaMA3-8B. The right block varies the backbone with a fixed collaboration structure.
>
> | Method               | 3 agents | 5 agents | 7 agents |-| LLaMA3-8B | LLaMA3-3B | Qwen-3B |
> |----------------------|----------|----------|----------|-|-----------|-----------|---------|
> | Central-Gen          | 0.8393   | 0.8638   | 0.8669   |-| 0.8393    | 0.7961    | 0.8832  |
> | Central-Gen+SC       | 0.8467   | 0.8853   | 0.8937   |-| 0.8467    | 0.8226    | 0.8916  |
> | Central-Gen+Select   | 0.8703   | 0.9032   | 0.8992   |-| 0.8703    | 0.8153    | 0.9083  |
>
>
> ---
>
> **Response to Q3 – Transfer beyond math and weaker end-state verifiability**
>
> We appreciate the question about transfer beyond math-style tasks with strong end-state verifiability. While we have not yet run explicit tool-use or long-horizon planning benchmarks in this work, we took a step in this direction by adding experiments on GPQA, a difficult expert-level scientific QA dataset where questions span biology, physics, and chemistry and are designed to be “Google-proof.” Compared with math word problems, GPQA has weaker and noisier end-state verifiability: there is a multiple-choice label, but the reasoning chains are long, cross-domain, and not easily checkable by symbolic tools.
>
> As reported in **Table 17**, MAESTRO and CLPO transfer well to this regime. Multi-agent collaboration already improves over single-agent prompting (Pure 0.2677, CoT 0.2828, SC 0.3081, GPTSwarm 0.3434), and MAESTRO in its prompt-only form reaches 0.3636. RL tuning then brings a further gain, with MAESTRO+GRPO at 0.3788 and MAESTRO+CLPO achieving 0.3940, about five points above the strongest baseline. This indicates that CLPO’s rationale-aware ranking is useful even when the final answer is not a numeric solution with an easily verifiable derivation, but a high-level scientific judgment where the model must select reliable rationales from a noisy candidate pool. Extending this rationale-ranking framework to tool-using and planning tasks, where rewards may come from softer or delayed signals (for example, environment feedback instead of exact answers), is a natural next step that we view as promising future work.
>
> **Table 17**: Comparison with RL-tuned multi-agent baselines on GSM8K, AMC, MMLU, and HumanEval. All methods use the same LLaMA-3-8B backbone and evaluation protocol.
>
> | Method   | GSM8K        | AMC          | MMLU         | HumanEval      |
> |----------|--------------|--------------|--------------|----------------|
> | ReMA     | 0.8719       | 0.2048       | 0.7125       | 0.6463         |
> | MAS-GPT  | 0.8635       | 0.2771       | 0.7096       | **0.6829**     |
> | MAESTRO  | **0.8933**   | **0.2852**   | **0.7238**   | 0.6687         |
>
> ---
>
> **We hope that these clarifications and new results address your main questions and provide a clearer picture of our approach. We truly welcome any further comments you may have and are ready to perform additional analyses if needed.**

---

> > ### Comment · Reviewer_d9LG · 2025-11-28
> >
> > I thank the authors for the comprehensive rebuttal. The addition of the RL-based baselines (MAS-GPT, ReMA) and the empirical validation of the coverage-identification theory (specifically tracking $p_t$ and $q_t$ during training) have effectively addressed my primary concerns. The results on GPQA further demonstrate the method's potential.

---

> > > ### Author Response · Authors · 2025-11-28
> > >
> > > **We sincerely thank you** for the thoughtful follow-up and for taking the time to read our revised experiments and analysis. We are glad that the additional results helped address your main concerns. The suggestions and comments you have provided have been extremely helpful in improving and refining our work.
> > >
> > > **If there are any remaining points** where further clarification or additional analysis could strengthen your confidence in the work, we would be very happy to provide it. **If it is possible** for our work to receive a higher evaluation, we would be deeply grateful. **We sincerely appreciate** your careful evaluation and constructive feedback throughout the review process.

---

### Official Review · Reviewer_VEx9 · 2025-11-01

**Soundness:** 4
**Presentation:** 4
**Contribution:** 3
**Rating:** 6
**Confidence:** 4

**Summary:**

This paper introduces Maestro, a technique for using a multi-agent LLM setup to improve performance on a range of benchmarks. The paper’s method contributes (1) a method for enabling divergence of solutions considered followed by convergence to a final solution, and (2) a modified RL technique for fine-tuning such approaches. They demonstrate superior performance on a range of benchmarks and perform a range of ablation and analysis experiments.

**Strengths:**

I overall find this paper fairly strong.

Originality: both the multi-agent framework, sans RL, and the RL are to my knowledge novel contributions. Further, I think the way the method is constructed is principled in a way that has clear relation to and interpretable differences from prior work — it’s a natural but still creative addition.

Quality: the experiments are extensive. The headline comparisons are good and the analysis experiments help answer a range of natural questions (how does it scale with number of agents? How does it compare to natural baselines for building consensus?). The performance decomposition into coverage and identification rates does a nice job of tying the experiments to methodological considerations introduced earlier.

Clarity: I generally found this work quite clear.

Significance: the results appear to be quite strong, with uniform improvement across settings and evidence that the method’s components all help.

**Weaknesses:**

Part of this stems from a question: I’d like to better understand how you control for inference budget. The appendix states that you match for collaboration budget (rounds, agents, and generations). Can one match all three, for all baselines? If so, how? Related, I’m trying to contextualize the results of Fig 2 with Fig 1: if I’m understanding correctly, the difference between SC and Central-Gen+SC is that Central-Gen+SC allows the convergence stage access to all of the divergent stage generations, then it gets to generate a number of candidates on its own (is that right?). Given that this seems to perform much better to SC…is the computational budget matched? Obviously computational budget matching would be helpful to consider here, given how some baselines might scale.

Related to this, it’s interesting that there appears to be degradation after a point in scaling agents, samples, and rounds. The intuition for this makes sense, but at the same time, suggests that this scales less well than naive baselines.

I also wish that some of the experiments (namely the Tab 2 and 3, in particular) were run on benchmarks that are a bit less saturated.

**Questions:**

I’d largely like help with understanding how baselines were compute matched, and my other scaling questions above.

---

> ### Author Response · Authors · 2025-11-26
> **Reply to Reviewer VEx9**
>
> We sincerely appreciate your careful and insightful review of our work. We have carefully reflected on your comments and carried out targeted analyses and experiments in response. We hope that the replies below can clarify our methodology and address your main concerns.
>
> ---
>
> **Response to W1 – On compute cost and budget control**
> Please see our reply to Reviewer 3wvd in the section **“Response to W4&5 – Compute cost, FLOP budget, and tradeoffs.”**
>
> ---
>
> **Response to W2 – Central-Gen+SC vs SC and compute budget**
>
> We appreciate your attention to the difference between Central-Gen+SC and SC, and **your understanding is correct**. Relative to standard self-consistency (SC), Central-Gen+SC augments the procedure with our Central-Gen framework. In SC, a single policy samples multiple independent solutions and then aggregates them. In Central-Gen+SC, we first run the divergent multi-agent stage to obtain a pool of candidate rationales. The central agent then conditions on this pool and generates additional candidates before applying self-consistency. This extra convergence stage naturally increases both the coverage of the candidate space and the total number of tokens.
>
> Given that computational budget is an important consideration, we analyze this aspect explicitly. By construction, Central-Gen widens Phase-1 exploration, so its token usage is necessarily higher than that of plain SC. A detailed token-cost analysis is provided in our reply to Reviewer 3wvd in the section “Response to W4&5 – Compute cost, FLOP budget, and tradeoffs,” where we report average input, output, and total tokens across all methods. The main observations are as follows. Our divergence–convergence design does incur additional token cost, but **increased tokens alone do not explain performance differences**. For example, strong single-agent SC already consumes a large number of tokens yet still underperforms the best multi-agent methods, indicating that **the collaboration pattern itself plays a crucial role**. Moreover, when we introduce a unified early-stopping strategy for all collaborative baselines, **MAESTRO maintains a clear accuracy advantage while its average token budget becomes comparable to other strong methods**. This suggests that the gains primarily stem from a more effective coordination scheme rather than from raw compute.
>
> It is also important to clarify the role of Central-Gen+SC in our study. The intent of this variant is not mainly to serve as a compute-matched replacement for SC, but to compare two designs for the central agent under the same divergent exploration. One option is a generative central agent that produces new solutions on top of the candidate pool, as in Central-Gen+SC. The other is a purely selective central agent that chooses among existing candidates, as in Central-Select. In our experiments, Central-Select consistently outperforms Central-Gen+SC while using fewer tokens. The average total token cost per problem is 12458 for Central-Select and 15392 for Central-Gen+SC. This comparison is exactly why MAESTRO **adopts a central selection policy** rather than central generation: it achieves higher accuracy with lower token cost given the same divergent stage, directly addressing both effectiveness and efficiency.
>
> ---
>
> **Response to W3 – Scaling behavior, degradation, and comparison to naive baselines**
>
> Please see our response to Reviewer 3wvd in the section **“Response to Q1&2 – Effect of the number of agents and rounds.”**

---

> ### Author Response · Authors · 2025-11-26
>
> **Response to W4 – On saturated benchmarks and additional evaluations**
>
> We appreciate the concern that some of our main benchmarks, especially those in Tables 2 and 3, are **relatively saturated**. This is indeed an important limitation of widely used math, coding, and MMLU benchmarks.
>
> First, recent multi-agent and RL-based methods such as G-Designer, MAS-GPT, and MaAS typically report absolute gains on GSM8K and MMLU in the range of about **0.2% to 0.5%** over strong baselines, and often below 1%. In our own experiments, we also observed that achieving a stable and reproducible gain larger than 1% over a carefully tuned GRPO baseline is difficult on these datasets. The improvements we report are therefore consistent with what is commonly observed in this saturated regime rather than unusually small.
>
> Second, to probe the effect of MAESTRO and CLPO in a less saturated setting, we added experiments on **GPQA**, a difficult expert-level scientific QA benchmark. GPQA is designed to be “Google-proof” and exhibits a large gap between non-expert performance and PhD-level experts. Results are reported in **Table 18**. On GPQA, multi-agent collaboration improves over single-agent prompting, and MAESTRO in its prompt-only form already outperforms LLM-Debate, Dylan, and GPTSwarm. With RL tuning, MAESTRO plus GRPO reaches **0.3788** accuracy and MAESTRO plus CLPO achieves **0.3940**. The margin over the strongest baseline GPTSwarm is **about five percentage points**, which is substantial on such a challenging dataset and indicates that CLPO improves the ability to identify correct answers under difficult scientific questions, not only on saturated math benchmarks.
>
>
>
> **Table 18**: Accuracy on the GPQA scientific QA benchmark for single-agent and multi-agent methods, including MAESTRO with and without RL tuning.
>
> | Method               | GPQA   |
> |----------------------|--------|
> | Pure                 | 0.2677 |
> | CoT                  | 0.2828 |
> | SC                   | 0.3081 |
> | LLM-Debate           | 0.3383 |
> | Dylan                | 0.3232 |
> | GPTSwarm             | 0.3434 |
> | MAESTRO              | 0.3636 |
> | MAESTRO + GRPO       | 0.3788 |
> | **MAESTRO + CLPO**   | **0.3940** |
>
> Third, we also evaluate transfer to a much larger backbone, **Llama3-70B**, on GSM8K, AMC, and MMLU. The results in **Table 9** show that the same MAESTRO collaboration structure and CLPO objective continue to yield consistent improvements over both vanilla prompting and GRPO-style RL across all three tasks. For example, MAESTRO plus CLPO reaches 94.85 on GSM8K, 63.86 on AMC, and 80.64 on MMLU. Compared with the vanilla 70B model, this corresponds to gains of 3.57, 2.40, and 5.77 points respectively, and MAESTRO plus CLPO also outperforms the strongest multi-agent baselines. This cross-scale behavior suggests that the method **is not tuned to a particular small model** or a specific benchmark, but provides a robust recipe that continues to add value even when the base policy is already strong.
>
> ---
>
> **We hope that our responses have helped to clarify our intentions and resolve at least some of your concerns. If you have any further questions or suggestions, we would be very happy to provide additional analysis or experiments.**

---

### Official Review · Reviewer_uZoA · 2025-11-01

**Soundness:** 2
**Presentation:** 3
**Contribution:** 1
**Rating:** 4
**Confidence:** 4

**Summary:**

The paper proposes MAESTRO, a multi-agent framework that explicitly separates divergent exploration (many execution agents generate K candidate reason-answer pairs) from convergent synthesis (a central agent selects one candidate and broadcasts it to guide later rounds). The core learning contribution is Conditional Listwise Policy Optimization (CLPO), which disentangles the update on decision tokens (which candidate to endorse) from rationale tokens (why), combining a choice-focused policy-gradient term with a listwise ranking loss over justifications, plus KL and entropy regularization. The authors analyze coverage vs identification (pt, qt) and give a simple success bound over T rounds. They report gains over single-agent and multi-agent baselines across common STEM+coding benchmarks. they also compare central generation vs central selection, arguing selection identifies correct candidates more reliably.

**Strengths:**

- The motivation is clear and problem is well-framed. The divergent vs convergent tension in collaborative LLMs is well articulated, and MAESTRO maps directly onto that cognitive split. This makes the system design easy to reason about. The credit assignment issue has been a pain point in the field of LLM RL, and any attempt to solve this problem is an appreciated effort.
- The paper is well-written and well articulated.

**Weaknesses:**

- Claimed improvements of about 1% over GRPO on math/coding/MMLU (~2% on AMC) are too small to justify the substantially more contrived objective and orchestration. I find it hard to see these increments as significant improvement over GRPO.
- The compute parity are under-specified for both all baselines. t’s unclear whether baselines (incl. GRPO) operate under matched total sampling budget (agent count × K × rounds), identical decoding setups, and equal reference-KL constraints. Without strict budget parity and detailed token-/call-level accounting, comparisons risk favoring MAESTRO merely by extra sampling/coordination.

**Questions:**

- The framework rebroadcasts the endorsed candidate each round, although an $\epsilon$-mixture is used to retain some base-policy exploration, how does entropy collapse across rounds?
- How does $\epsilon$ affect training stability

---

> ### Author Response · Authors · 2025-11-26
> **Reply to Reviewer uZoA**
>
> We really appreciate your careful and insightful review of our work. We have carefully gone through your comments and conducted targeted analyses and experiments in response, and we hope that the following replies can alleviate your concerns and clarify our contributions.
>
> ---
>
> **Reply to Weaknesses:**
>
> **Response to W1 – Magnitude and significance of improvements over GRPO**
>
> Thank you for raising the concern about the **absolute size of the gains over GRPO**. We also see this as an important question and therefore conduct the following analysis and experiments.
>
> First, the widely used math, coding, and MMLU benchmarks are already in a **relatively high saturated regime**. Recent multi-agent and RL-based methods such as G-Designer, MAS-GPT, and MaAS typically report absolute gains on GSM8K and MMLU that are often in the range of about **0.2% to 0.5% points** over strong baselines, and in many cases below one percentage point. In our own experiments, we also observed that obtaining a stable and reproducible gain larger than 1% over a carefully tuned GRPO baseline is quite difficult on these datasets. The improvements we report, roughly one point on math, coding, and MMLU and around two points on AMC, are therefore consistent with what is commonly observed in this saturated regime rather than unusually small. Another reviewer (VEx9) explicitly highlighted that these benchmarks are close to saturation for current models, which further supports this interpretation.
>
> Second, to probe the effect of CLPO in a less saturated setting, we added experiments on **GPQA**, a difficult expert-level scientific QA dataset. GPQA is designed to be “Google-proof” and has a large gap between non-expert performance and PhD-level experts. Results are reported in **Table 18**. On this benchmark, multi-agent collaboration improves over single-agent prompting, and MAESTRO in its prompt-only form already outperforms LLM-Debate, Dylan, and GPTSwarm. RL tuning then provides a clear additional benefit. MAESTRO plus GRPO reaches 0.3788 accuracy, while MAESTRO plus CLPO attains 0.3940. The margin over the strongest baseline GPTSwarm is about five points. In this high-difficulty regime, such a gap is substantial and indicates that CLPO is not merely fitting noise on a specific math dataset but improves the ability to identify correct answers under challenging scientific questions.
>
> **Table 18**: Accuracy on the GPQA scientific QA benchmark for single-agent and multi-agent methods, including MAESTRO with and without RL tuning.
>
> | Method               | GPQA   |
> |----------------------|--------|
> | Pure                 | 0.2677 |
> | CoT                  | 0.2828 |
> | SC                   | 0.3081 |
> | LLM-Debate           | 0.3383 |
> | Dylan                | 0.3232 |
> | GPTSwarm             | 0.3434 |
> | MAESTRO              | 0.3636 |
> | MAESTRO + GRPO       | 0.3788 |
> | **MAESTRO + CLPO**   | **0.3940** |
>
>
> Third, we also evaluate transfer to a much larger backbone, **Llama3-70B**, on GSM8K, AMC, and MMLU. The results in **Table 9** show that the same MAESTRO structure and CLPO objective yield **consistent improvements over both vanilla prompting and GRPO-style RL across all three tasks**. For example, MAESTRO plus CLPO reaches 94.85 on GSM8K, 63.86 on AMC, and 80.64 on MMLU, improving over the vanilla 70B model by 3.57, 18.08, and 5.77 points respectively, and also outperforming the strongest multi-agent baselines. This cross-scale behavior suggests that the method is not tuned to a particular small model, but provides a robust recipe that continues to add value even when the base policy is strong.
>
>
> **Table 9**: Performance (%) of **Llama3-70B** on GSM8K, AMC, and MMLU under different prompting and multi-agent collaboration strategies.
>
> | Method             | GSM8K | AMC   | MMLU  |
> |--------------------|:-----:|:-----:|:-----:|
> | Vanilla            | 91.28 | 45.78 | 74.87 |
> | CoT                | 91.81 | 49.40 | 75.72 |
> | SC                 | 92.57 | 54.21 | 77.28 |
> | LLM-Debate         | 92.87 | 57.83 | 78.08 |
> | GPTSwarm           | 93.18 | 54.22 | 78.35 |
> | DYLAN              | 92.27 | 56.63 | 77.18 |
> | MAESTRO            | 93.86 | 59.04 | 79.06 |
> | MAESTRO + GRPO     | 94.16 | 61.44 | 79.73 |
> | **MAESTRO + CLPO** | **94.85** | **63.86** | **80.64** |
>
> While the absolute gains over GRPO on some saturated benchmarks are numerically modest, they are **consistent across diverse datasets and model sizes**, and they become clearly pronounced on harder settings such as AMC and GPQA. We view CLPO not only as an incremental tweak to GRPO, but as a principled way to disentangle choice and rationale supervision for the central selector within MAESTRO. The experimental results show that this additional structure translates into reliable performance improvements rather than being an overly contrived objective with negligible effect.

---

> ### Author Response · Authors · 2025-11-26
>
> **Response to W2 – On compute cost and budget control**
>
> Please see our reply to **Reviewer 3wvd in the section “Response to W4&5 – Compute cost, FLOP budget, and tradeoffs.”**
>
> ---
> **Reply to questions:**
>
> **Response to Q1&2 – Rebroadcast, exploration mixture, and collapse concerns**
>
> We appreciate your question about how rebroadcast affect diversity across rounds and the stability of CLPO training. We address both aspects below.
>
> First, regarding entropy and potential collapse across rounds. In MAESTRO each execution agent conditions on a mixture of two inputs in every round. With probability $1 - \varepsilon$ it receives the rebroadcast endorsed candidate in its context and with probability $\varepsilon$ it instead follows the base prompt without rebroadcast. In all reported experiments we fix $\varepsilon$ to $0.1$ and do not tune it per task. This mixture means that even as the endorsed candidate is propagated across rounds, there is always a non-trivial fraction of trajectories that are generated purely under the base policy, which prevents the distribution from collapsing to a single deterministic pattern.
>
> To investigate this question, we conducted a diversity study, which shows that this mechanism does not lead to collapse. As reported in **Table 5** (see reply to Reviewer 3wvd or Appendix D), MAESTRO produces the highest coverage together with substantially more distinct final answers and high inconsistency rates on both datasets, indicating persistent cross-agent disagreement rather than convergence to one shared answer. The ablations in Tables 6 and 7 further show that increasing the number of agents or the per-agent sampling depth increases coverage and distinct answer counts while inconsistency remains high. Taken together, these results suggest that even after multiple rounds of rebroadcast, MAESTRO maintains a rich candidate distribution instead of collapsing to a single mode. In other words, entropy is intentionally reduced enough to enable coordination but remains bounded away from zero due to the $\varepsilon$-mixture and the diversity knobs in Phase 1.
>
> Second, regarding the effect of this design on training stability. In CLPO the only new scalar hyperparameter is the rationale ranking weight $\lambda_{\text{rank}}$. The KL and entropy terms use standard RLHF values and are fixed in all experiments. We conducted a sensitivity study of $\lambda_{\text{rank}}$ on GSM8K, AMC, and MMLU, and **Table 7** (see reply to Reviewer 3wvd or Appendix D) shows smooth performance curves across $\lambda_{\text{rank}}$ in the range from $0$ to $1.0$. Moderate values around $0.3$ to $0.5$ consistently improve over GRPO and even extreme values do not cause instability. Combined with the fact that a fixed $\varepsilon = 0.1$ works across all tasks and model sizes without retuning, this indicates that the training dynamics of CLPO under the $\varepsilon$-mixture are stable rather than brittle. In future work it would be interesting to make $\varepsilon$ adaptive, for example based on uncertainty, but in the current setting a simple fixed mixture already yields stable training and non-collapsed multi-round behavior.
>
> ---
>
> **We hope the above responses have addressed your questions. If you have any further comments or suggestions, please do not hesitate to let us know.** We would be very happy to provide additional analysis or experiments. If possible, we would also kindly ask you to reconsider your evaluation of our work. We are deeply grateful for your time, feedback, and support.

---

### Official Review · Reviewer_3wvd · 2025-11-01

**Soundness:** 2
**Presentation:** 2
**Contribution:** 3
**Rating:** 4
**Confidence:** 4

**Summary:**

Multi-agent systems can be more powerful than single LLM agents alone. Yet, they are difficult to get right because of difficulties in diverging exploration of the solution space by different agents, and convergent synthesis of the found solutions. Particularly, credit assignment is hard for such systems. Therefore, they propose Multi-Agent Exploration Synthesis framework through Role Orchestrating (MAESTRO) in combination with Conditional Listwise Policy Optimization (CLPO) to enable effective multi-agent systems. Their experiments on commonly used reasoning tasks show gains compared to previous approaches.

**Strengths:**

1. The authors address an important problem in multi-agent LLM systems, that is, divergent exploration followed by convergent synthesis of the explored solution space.
2. The proposed MAESTRO framework seems sound and nicely decomposes exploration and convergence.
3. The proposed CLPO RL objective effectively decomposes decisions and reasons to enable more effective credit-assignment.
4. They provide interesting analyses in section 4.2 on the effect of collaboration mechanisms, parts of the CLPO loss, and the size of the agent population.

**Weaknesses:**

1. It is unclear how diverse the generated solutions in Phase 1 really are. Is there any way to enforce more diversity in this phase?
2. The proposed CLPO objective (despite good motivation) consists of 4 loss terms with individual weights. However, there is no analysis on the sensitivity of the weighting terms of the proposed components. This would be valuable for the reader to better understand the difficulty of tuning CLPO.
3. For RL tuning, the paper primarily focuses on the small-scale models (Llama3 3/8B and Qwen 3B/7B). It is unclear how well their recipes transfer to larger-scale models. I understand the difficulty of doing that, given that MAESTRO requires many more rollouts due to the multi-agent setup. However, this is important to understand better.
4. MAESTRO requires many more rollouts due to the multi-agent setup. However, there is no wall-clock/FLOP analysis in the paper compared to using only a single agent. This would be important for the reader to understand better the tradeoffs between using multiple agents and using a single agent rollout. Generally, I would like to see a FLOP budget or wall-clock time column for all methods in Tables 1,2, and 3. It is possible that the increased scores and the result of the increased compute spent.
5. For self-consistency, debate, etc., did you control for the FLOP budget? I.e., do these methods spend a comparable amount of compute to MAESTRO, or is this way less? I think this is an important comparison, especially because improvement margins are slim for some tasks.
6. I appreciate the ablation studies in Sections 4.2 and 4.3. Interestingly, performance declines as more agents are added, more samples are generated per agent, and more rounds are conducted. This raises concerns regarding the scalability of your recipe. I would be interested in your thoughts on this.

Minor:
- In section 3, both the question and the identification probability are denoted as $q$.

**Questions:**

Some questions are listed in the Weakness section. Here are some more questions:
1. Why does the performance on AMC, GSM8K decline as the number of agents increases? What would happen for more than 5 agents? Does this result limit the scalability of your method? How might this change for larger-scale models?
3. For the number of collaboration rounds, this effect is even more pronounced (Figure 6 in the appendix). What is your intuition why that’s the case, and how can it be overcome?

---

> ### Author Response · Authors · 2025-11-26
> **Reply to Reviewer 3wvd**
>
> We sincerely thank you for your careful and thoughtful review of our work. Your comments are highly valuable to us. For the concerns you raised, we hope the following responses can provide effective clarification.
>
> ---
>
> **Reply to Weaknesses:**
>
> **Response to W1 – Diversity of Phase-1 Solutions and How to Enforce It**
>
> We appreciate your attention to the diversity of solutions generated in Phase 1, this is an important aspect of multi-agent collaboration quality. To more clearly characterize how diverse our Phase-1 candidates are, and how this diversity can be further enhanced, we have conducted additional analyses and ablations as follows.
>
> **(1) How diverse are the Phase-1 solutions?**
> We added a dedicated diversity study that compares MAESTRO against strong multi-agent baselines (LLM-Debate, Dylan, GPTSwarm) under the same backbone (LLaMA3-8B) and decoding configuration. For each problem and method, we analyze the candidate slate produced in the final collaboration round and report three metrics:
>
> - **Coverage rate:** fraction of problems whose slate contains at least one correct final answer.
> - **Average distinct answers:** average number of unique final answers per slate after deduplication.
> - **Inconsistency rate:** fraction of problems where different agents produce at least two distinct final answers (i.e., explicit cross-agent disagreement).
>
> On both datasets, MAESTRO attains the **highest coverage** (e.g., 0.2892 vs. 0.1928 on AMC, 0.9325 vs. 0.8522 on GSM8K compared to the best baseline) while also generating **substantially more distinct final answers** per problem (7.55 vs. 2.86 on AMC and 2.92 vs. 2.05 on GSM8K relative to LLM-Debate), as summarized in **Table 5**. The inconsistency rate is similarly high (0.98 on AMC and 0.80 on GSM8K), indicating that agents frequently disagree and explore different hypotheses rather than collapsing to a single shared solution. We emphasize that coverage is the primary quality indicator, while average distinct answers and inconsistency are descriptive statistics that confirm the candidate pool is rich and non-degenerate rather than narrowly concentrated.
>
> **Table 5**: Phase-1 diversity metrics across multi-agent methods on AMC and GSM8K.  *Coverage rate* is the fraction of problems whose candidate slate contains at least one correct answer.  *Avg. distinct answers* is the average number of unique final answers per slate.  *Inconsistency rate* is the fraction of problems where agents produce at least two different final answers.
>
> | Dataset | Method      | Coverage rate | Avg. distinct answers | Inconsistency rate |
> |--------|------------|--------------:|-----------------------:|-------------------:|
> | AMC     | LLM-Debate | 0.1928        | 2.86                   | 0.96               |
> | AMC        | Dylan      | 0.1807        | 2.77                   | 0.93               |
> | AMC        | GPTSwarm   | 0.1325        | 2.48                   | 0.83               |
> | AMC        | **MAESTRO**| **0.2892**    | 7.55                   | 0.98               |
> | GSM8K   | LLM-Debate | 0.8522        | 2.05                   | 0.68               |
> | GSM8K        | Dylan      | 0.8264        | 1.41                   | 0.31               |
> | GSM8K        | GPTSwarm   | 0.8385        | 1.64                   | 0.46               |
> | GSM8K        | **MAESTRO**| **0.9325**    | 2.92                   | 0.80               |

---

> ### Author Response · Authors · 2025-11-26
>
> **(2) Can diversity be explicitly enforced or increased?**
> Yes. We further show that MAESTRO exposes simple, interpretable knobs to trade additional compute for higher diversity:
>
> - **Varying the number of execution agents $N$ (width):** Increasing $N$ with fixed $K=3$ consistently improves coverage and distinct answer counts (see **Table 6**). For example, on AMC the coverage rate increases from 0.2892 (2 agents) to 0.3735 (5 agents), while the average number of distinct answers grows from 5.20 to 11.98. On GSM8K, coverage rises from 0.9083 to 0.9409 and distinct answers from 2.84 to 3.97 as $N$ increases, with inconsistency rates remaining high. This indicates that additional agents contribute genuinely different hypotheses rather than redundant copies.
>
> - **Varying the per-agent sampling depth $K$ (depth):** Increasing $K$ with fixed $N=3$ yields a similar pattern (see **Table 7**). On AMC, coverage improves from 0.2651 ($K=2$) to 0.3855 ($K=5$), and average distinct answers from 5.10 to 12.47; on GSM8K, coverage rises from 0.9022 to 0.9416 with steadily increasing distinct answer counts. Inconsistency rates stay in a high regime across settings, reflecting persistent disagreement and thus diversity in the candidate pool.
>
>
> **Table 6**: Effect of the number of agents on Phase-1 diversity metrics for MAESTRO on AMC and GSM8K.
>
> | Dataset | Agents | Coverage rate | Avg. distinct answers | Inconsistency rate |
> |--------|:------:|--------------:|-----------------------:|-------------------:|
> | AMC    |   2    | 0.2892        | 5.20                   | 0.98               |
> | AMC    |   3    | 0.3494        | 7.55                   | 0.98               |
> | AMC    |   4    | 0.3614        | 9.69                   | 0.99               |
> | AMC    |   5    | **0.3735**    | 11.98                  | 0.94               |
> | GSM8K  |   2    | 0.9083        | 2.84                   | 0.75               |
> | GSM8K  |   3    | 0.9325        | 2.92                   | 0.80               |
> | GSM8K  |   4    | 0.9386        | 3.33                   | 0.82               |
> | GSM8K  |   5    | **0.9409**    | 3.97                   | 0.87               |
>
> **Table 7**: Effect of per-agent sampling depth K on Phase-1 diversity metrics for MAESTRO on AMC and GSM8K.
>
> | Dataset | K | Coverage rate | Avg. distinct answers | Inconsistency rate |
> |--------|:-:|--------------:|-----------------------:|-------------------:|
> | AMC    | 2 | 0.2651        | 5.10                   | 0.94               |
> | AMC    | 3 | 0.3494        | 7.55                   | 0.98               |
> | AMC    | 4 | 0.3735        | 9.78                   | 0.98               |
> | AMC    | 5 | **0.3855**    | 12.47                  | 0.99               |
> | GSM8K  | 2 | 0.9022        | 2.24                   | 0.64               |
> | GSM8K  | 3 | 0.9325        | 2.92                   | 0.80               |
> | GSM8K  | 4 | 0.9378        | 3.38                   | 0.80               |
> | GSM8K  | 5 | **0.9416**    | 3.85                   | 0.87               |
>
> These ablations demonstrate that Phase-1 diversity in MAESTRO is not an opaque by-product of decoding, but can be **systematically increased** by adjusting $N$ and $K$. We included these new experiments and a discussion of these diversity controls in the revised manuscript.

---

> ### Author Response · Authors · 2025-11-26
>
> **Response to W2 – Sensitivity of CLPO weighting terms**
>
> Thank you for raising the question about the potential tuning difficulty of CLPO. Although the objective is written with four terms, in practice only one of them introduces a new source of sensitivity.
>
> CLPO is composed of a choice loss, a rationale ranking loss, a KL regularizer, and an entropy bonus. The KL and entropy components follow standard RLHF and GRPO practice and we keep their weights fixed in all experiments without tuning, with $\lambda_{\text{KL}} = 0.1$ and $\lambda_{\text{entropy}} = 0.01$. The only additional coefficient specific to CLPO is the rationale ranking weight $\lambda_{\text{rank}}$. When $\lambda_{\text{rank}} = 0$, the objective reduces to a GRPO style training procedure without explicit ranking supervision.
>
> To address the concern directly, we conducted a sensitivity study of $\lambda_{\text{rank}}$ on GSM8K, AMC, and MMLU, and report end to end accuracy in Table 8. For each dataset we train with $\lambda_{\text{rank}} \in \\{0, 0.1, 0.3, 0.5, 1.0\\}$ under identical model and optimization settings. On GSM8K and AMC, moving from $\lambda_{\text{rank}} = 0$ to moderate values around 0.3 to 0.5 consistently improves performance over the GRPO style baseline. For GSM8K, accuracy increases from 0.8848 to a peak of 0.9030 at $\lambda_{\text{rank}} = 0.5$. For AMC, accuracy increases from 0.2530 to 0.3133 at the same setting. Even at $\lambda_{\text{rank}} = 1.0$, performance remains close to or better than the $\lambda_{\text{rank}} = 0$ case. On MMLU, accuracy rises smoothly from 0.7132 at $\lambda_{\text{rank}} = 0$ to 0.7238 at $\lambda_{\text{rank}} = 1.0$, with only small variation across the range.
>
> Taken together, these results indicate that CLPO is **not highly sensitive to the exact choice** of $\lambda_{\text{rank}}$. A simple default in a moderate range, for example between 0.3 and 0.5, is sufficient to obtain stable and consistent gains over GRPO style training without extensive hyper-parameter tuning.
>
> **Table 8**: Accuracy of MAESTRO+CLPO on GSM8K, AMC, and MMLU under different values of the rationale ranking weight λ_rank.
>
> | Dataset | 0.0   | 0.1   | 0.3   | 0.5        | 1.0   |
> |---------|:-----:|:-----:|:-----:|:----------:|:-----:|
> | AMC     | 0.2530 | 0.2771 | 0.2892 | **0.3133** | 0.2852 |
> | GSM8K   | 0.8848 | 0.8893 | 0.8954 | **0.9030** | 0.8933 |
> | MMLU    | 0.7132 | 0.7169 | 0.7201 | 0.7215     | **0.7238** |
>
> ---
>
> **Response to W3 – Transfer of RL tuning to larger models**
>
> We agree that it is important to understand whether our collaboration and RL tuning recipes transfer beyond small and mid scale models. In the original submission we focused on 3B–8B backbones, both **to fairly compare with related works** that also use similarly sized models and to keep the rollout cost manageable in the multi-agent setting. To address this concern more directly, we have added experiments with **a larger backbone, Llama3-70B**, on three representative benchmarks: GSM8K, AMC, and MMLU. All methods share the same 70B base model and decoding configuration. For MAESTRO we keep the collaboration structure (number of agents and rounds) and CLPO settings the same as in the smaller scale experiments, and only adjust batch size to fit memory.
>
> With this setup, MAESTRO and its RL tuned variants continue to provide consistent gains over both single agent prompting and other multi-agent baselines, and the results are reported in Table 9. Moving from the Vanilla single agent 70B model to multi-agent methods already improves accuracy. MAESTRO in its prompt only form further improves on these baselines on all three datasets. Adding RL tuning gives an additional and stable boost. MAESTRO plus GRPO improves GSM8K and AMC, and MAESTRO plus CLPO achieves the best performance overall, with 94.85 on GSM8K, 63.86 on AMC, and 80.64 on MMLU. These results show that the collaboration pattern and CLPO objective do not saturate when the backbone is strong. They transfer to a 70B model without special re-tuning, and still yield non-trivial gains in the high accuracy regime.
>
> **Table 9**: Performance (%) of **Llama3-70B** on GSM8K, AMC, and MMLU under different prompting and multi-agent collaboration strategies.
>
> | Method             | GSM8K | AMC   | MMLU  |
> |--------------------|:-----:|:-----:|:-----:|
> | Vanilla            | 91.28 | 45.78 | 74.87 |
> | CoT                | 91.81 | 49.40 | 75.72 |
> | SC                 | 92.57 | 54.21 | 77.28 |
> | LLM-Debate         | 92.87 | 57.83 | 78.08 |
> | GPTSwarm           | 93.18 | 54.22 | 78.35 |
> | DYLAN              | 92.27 | 56.63 | 77.18 |
> | MAESTRO            | 93.86 | 59.04 | 79.06 |
> | MAESTRO + GRPO     | 94.16 | 61.44 | 79.73 |
> | **MAESTRO + CLPO** | **94.85** | **63.86** | **80.64** |

---

> ### Author Response · Authors · 2025-11-26
>
> **Response to W4&5 – Compute cost, FLOP budget, and tradeoffs**
>
> Thank you for raising the questions about compute cost and FLOP budget. In practice it is difficult to enforce exactly the same FLOP budget across methods, because their interaction patterns are structurally different. In our setting, different methods inherently follow different interaction patterns, which makes strict FLOP matching nontrivial. Following prior work on multi-agent collaboration, we therefore adopt a unified experimental protocol in the main results, where all methods use the same backbone, decoding configuration, number of agents, and number of collaboration rounds. On top of this common setup, we now provide both a **token-based cost analysis** and a **wall-clock analysis** as practical, hardware-agnostic proxies for FLOPs.
>
> **First**, we measure average input tokens, output tokens, and total **tokens per problem** for all methods. The detailed numbers are given in **Table 10**. MAESTRO indeed uses the largest token budget when all planned agents, rounds, and samples are used. For example, on GSM8K the average total tokens per problem are 12,458 for MAESTRO, compared with 7,627 for GPTSwarm and 8,758 for LLM-Debate. On AMC the numbers are 24,282 for MAESTRO, 17,827 for LLM-Debate and 12,952 for GPTSwarm. At the same time, token usage does not correlate monotonically with accuracy. Single-agent self-consistency already consumes many tokens, 1,943 on GSM8K and 3,944 on AMC, but still performs worse than the best multi-agent baselines. This suggests that the gains of MAESTRO are not simply a consequence of spending more tokens, but of how the additional compute is organized into Phase-1 exploration and centralized selection. The higher cost mainly reflects a deliberate choice to allocate more samples to increase candidate diversity, which is central to the design of the framework.
>
> **Second**, motivated by the concern that full-budget MAESTRO is relatively expensive, we ask whether MAESTRO can operate in a more cost-aware regime while retaining most of its benefits. To study this, we introduce an **early-stopping strategy** (for frameworks without a central agent, stopping once the majority vote over agents’ answers converges across rounds; for frameworks with a central agent, stopping once the central selector’s chosen candidate stabilizes across consecutive rounds) and apply it uniformly to all collaborative methods, including MAESTRO, LLM-Debate, Dylan and GPTSwarm, and then recompute both solve rate and token cost. A run can terminate before the maximum number of rounds when a simple consensus or correctness heuristic is satisfied. Results with early-stopping appear in **Table 11**. On GSM8K, MAESTRO reaches a solve rate of 0.8789 with an average of 8,172 tokens per problem. GPTSwarm reaches 0.8372 with 6,385 tokens, and LLM-Debate reaches 0.8235 with 9,847 tokens. On AMC, MAESTRO reaches 0.2661 with 12,395 tokens, while Dylan reaches 0.1968 with 11,525 tokens and LLM-Debate reaches 0.1738 with 14,473 tokens. In this regime, the token cost of MAESTRO is in the same order of magnitude as other strong multi-agent methods. Taken together, Table 10 and Table 11 show that **MAESTRO supports a flexible token-level compute tradeoff**. The full-budget configuration illustrates the upper bound when compute is plentiful, and the early-stopping configuration shows that most of the accuracy gains can be preserved under a much tighter token budget.
>
> **Third**, to complement the token-based view with a more direct notion of runtime, we report **wall-clock efficiency** in **Table 12** and visualize the **compute–performance frontier in  Appendix D-Figure 7**. We measure the average seconds per problem and the corresponding throughput (problems per second) under a shared hardware and batch-size setting. As expected, the single-agent Vanilla model is the fastest (0.26 s/problem, 3.79 problems/s) but also has the lowest solve rate (0.73), while self-consistency improves accuracy to 0.81 at roughly a 3× increase in wall-clock cost. Multi-agent baselines such as Dylan and GPTSwarm require around 3.9 s/problem yet remain in the 0.82–0.85 accuracy range, and LLM-Debate sits in between at 2.05 s/problem and 0.8352 solve rate. In contrast, MAESTRO with two agents already attains a solve rate of 0.8693 at 2.57 s/problem, comparable in cost to LLM-Debate but with a clear accuracy gain. As we increase the number of agents, MAESTRO traces a smooth compute–performance frontier: the 3-agent configuration reaches 0.8933 solve rate at 4.26 s/problem, and the 4-agent configuration yields the highest accuracy of 0.9037 at 5.82 s/problem. Thus, **even when measured in wall-clock time rather than tokens, MAESTRO offers better accuracy–efficiency trade-offs than prior multi-agent frameworks**, and scaling collaboration allows practitioners to choose points along a controllable compute–performance curve rather than simply spending unbounded additional compute.

---

> ### Author Response · Authors · 2025-11-26
>
> **Table 10**: Average token usage per problem. For each method we report the average input tokens, average output tokens, and their sum as the average total tokens per problem.
>
> | Dataset | Method     | Avg. input tokens | Avg. output tokens | Avg. total tokens |
> |--------|-----------|-------------------|--------------------|-------------------|
> | GSM8K  | Vanilla   | 114               | 182                | 296               |
> | GSM8K  | SC        | 119               | 1,824              | 1,943             |
> | GSM8K  | LLM-Debate| 6,974             | 1,784              | 8,758             |
> | GSM8K  | Dylan     | 4,135             | 3,476              | 7,611             |
> | GSM8K  | GPTSwarm  | 5,525             | 2,102              | 7,627             |
> | GSM8K  | MAESTRO   | 7,155             | 5,303              | 12,458            |
> |    |   |               |                |               |
> | AMC    | Vanilla   | 145               | 449                | 593               |
> | AMC    | SC        | 150               | 3,794              | 3,944             |
> | AMC    | LLM-Debate| 13,968            | 3,859              | 17,827            |
> | AMC    | Dylan     | 8,208             | 3,317              | 11,525            |
> | AMC    | GPTSwarm  | 9,291             | 3,660              | 12,952            |
> | AMC    | MAESTRO   | 14,221            | 10,061             | 24,282            |
>
>
> **Table 11**: Solve rate and average total tokens per problem with early-stopping strategy.
>
> | Method    | GSM8K Solve rate | GSM8K Avg. total tokens | AMC Solve rate | AMC Avg. total tokens |
> |-----------|------------------|-------------------------|----------------|-----------------------|
> | Vanilla   | 0.7276           | 296                     | 0.0803         | 593                   |
> | SC        | 0.8079           | 1,943                   | 0.1165         | 3,944                 |
> | LLM-Debate| 0.8235           | 9,847                   | 0.1738         | 14,473                |
> | Dylan     | 0.8203           | 7,611                   | 0.1968         | 11,525                |
> | GPTSwarm  | 0.8372           | 6,385                   | 0.1284         | 9,362                 |
> | MAESTRO   | **0.8789**       | 8,172                   | **0.2661**     | 12,395                |
>
> **Table 12**: Wall-clock efficiency and solve rate on GSM8K. We report the average seconds per problem, the corresponding throughput (problems per second), and the solve rate for all methods, measured on the same hardware and batch size.
> | Method               | Sec / problem | Problems / sec | Solve rate |
> |----------------------|---------------|----------------|-----------:|
> | Vanilla              | 0.26          | 3.79           | 0.7276     |
> | Self-Consistency (SC)| 0.72          | 1.40           | 0.8079     |
> | Dylan                | 3.91          | 0.26           | 0.8203     |
> | LLM-Debate           | 2.05          | 0.49           | 0.8352     |
> | GPTSwarm             | 3.89          | 0.26           | 0.8489     |
> | MAESTRO (2 agents)   | 2.57          | 0.39           | 0.8693     |
> | MAESTRO (3 agents)   | 4.26          | 0.24           | 0.8933     |
> | MAESTRO (4 agents)   | 5.82          | 0.17           | **0.9037**     |

---

> ### Author Response · Authors · 2025-11-26
>
> **Response to W6 – On scalability with respect to the number of agents and rounds**
>
> Please see our responses to Questions 1 and 2 below.
>
> ---
>
> **Reply to Questions:**
>
> **Response to Q1 – Effect of the number of agents and scalability**
>
> We appreciate your question about **how performance scales with the number of agents** and whether the trends on AMC and GSM8K point to a **fundamental limitation of the method**. In the revised version we extend the sweep to six and seven agents for all multi-agent baselines, and report the full results in **Table 13**. Across both datasets, all multi-agent methods, including LLM-Debate, DYLAN, GPTSwarm, and MAESTRO, show a similar pattern. Moving from two agents to a moderate ensemble size clearly helps, after which performance enters a high but non-monotonic regime. For MAESTRO, accuracy improves when we go from two to around four or five agents, and then stays in a narrow band with small fluctuations as we increase to six and seven agents. The other baselines show the same qualitative behaviour. This suggests that additional agents are beneficial up to a certain width, and that beyond this point the system essentially **saturates rather than collapsing** as more agents are added.
>
> We believe this behaviour is driven by **two main factors** that are typical for math and reasoning tasks. **On the task side**, once the ensemble already achieves high coverage and identification, extra agents mostly add redundant or noisy rationales. The Phase-1 slate grows longer, the central policy must process much longer context, and long-context reasoning becomes harder. At the same time, more agents increase the chance that strongly worded but incorrect rationales appear in the slate, which can confuse the selector instead of helping it. **On the optimization side**, CLPO is trained to maximize end-to-end solve rate under a fixed collaboration pattern, and the learning signal is aggregated across agents and rounds. The objective does not enforce that every increase in the number of agents must yield a strictly better policy. In this light, the observed plateau with mild oscillations is consistent with a **coverage–noise trade-off**. Early increases in width raise coverage and sharpen identification, while later increases mainly inject noisy or redundant information. In practice this does not limit scalability, because **MAESTRO works well with a modest number of agents and exposes an explicit knob to trade compute for accuracy**. For larger backbones with stronger base policies, such as LLaMA3-70B, high coverage is already achieved with fewer agents, so the useful operating regime shifts even more toward smaller ensembles rather than requiring ever larger teams.
>
> This does not limit scalability in practice, because the number of agents in MAESTRO is **an explicit knob rather than a hidden requirement**. The experiments indicate that **a small ensemble of three to five agents already captures most of the attainable gains**, and that adding more agents beyond this range gives **diminishing returns while increasing token cost**. For larger backbones such as LLaMA3-70B (Table 9), the base policy is stronger and needs fewer agents to reach high coverage, so we expect the useful operating regime to skew even more toward moderate widths. In our view this is a feature rather than a limitation: MAESTRO remains effective with a modest number of agents and exposes a controllable trade-off between compute and accuracy, rather than requiring ever larger ensembles to work well.
>
>
> **Table 13**: Impact of the number of agents on the solve rate of different multi-agent methods.
>
> | Dataset | Method     | 2 agents | 3 agents | 4 agents | 5 agents | 6 agents | 7 agents |
> |---------|-----------|----------|----------|----------|----------|----------|----------|
> | GSM8K   | LLM-Debate| 0.8241   | 0.8352   | 0.8309   | 0.8393   | 0.8286   | 0.8347   |
> | GSM8K   | DYLAN     | 0.8165   | 0.8203   | 0.8582   | 0.8521   | 0.8552   | 0.8628   |
> | GSM8K   | GPTSwarm  | 0.8067   | 0.8489   | 0.8628   | 0.8575   | 0.8514   | 0.8545   |
> | GSM8K   | **MAESTRO**   | **0.8693** | **0.8933** | **0.9037** | **0.9032** | **0.8961** | **0.8992** |
> | AMC     | LLM-Debate| 0.1807   | 0.1928   | 0.2289   | 0.2169   | 0.2048   | 0.2651   |
> | AMC     | DYLAN     | 0.1687   | 0.1968   | 0.2169   | 0.2410   | 0.2651   | 0.2538   |
> | AMC     | GPTSwarm  | 0.1566   | 0.1566   | 0.1446   | 0.1928   | 0.1566   | 0.1687   |
> | AMC     | **MAESTRO**   | **0.2530** | **0.2852** | **0.3052** | **0.2972** | **0.3133** | **0.3012** |

---

> ### Author Response · Authors · 2025-11-26
>
> **Response to Q2 – Effect of the number of collaboration rounds and mitigation**
>
> We appreciate your question about **how performance scales with the number of collaboration rounds**. Our detailed analysis is as follows.
>
> First, we extended the round ablation to all strong multi-agent baselines under the same backbone and decoding setup. The results are reported in **Table 14**. We observe that **non-monotonic trends are not unique to MAESTRO**. LLM Debate, Dylan, and GPTSwarm almost always follow a similar pattern. Performance improves when moving from two to a small number of rounds, then either plateaus or drops once the number of rounds is pushed further. **MAESTRO follows the same general shape but at a higher level**, for example on GSM8K we move from 0.8789 to 0.8933 at three rounds, then see a small decline to 0.8872 and 0.8842 at four and five rounds. This suggests that the effect reflects a broader property of iterative collaboration on these tasks, not only our specific framework.
>
> We believe that this phenomenon mainly stems from **two parts**. The first is about **the underlying tasks**. Mathematical and reasoning benchmarks are sensitive to spurious reasoning and numerical mistakes. As the number of rounds grows, agents operate on increasingly long and noisy contexts that combine multiple partial attempts and justifications. This makes the input harder to parse and increases the chance that a confident but wrong intermediate answer becomes the new focal point. In other words, more turns amplify both useful corrections and misleading cues, and beyond a certain point the latter starts to dominate. Similar diminishing returns appear when we increase the number of agents or the per agent sampling depth. A small increase in diversity helps the central selector, but a very large pool of noisy candidates makes it harder to identify the best one reliably.
>
> The second part concerns **the learning objectives themselves**. Prompt-based baselines such as debate and swarm rely on heuristic rules for how agents update their messages across rounds, without an explicit optimality condition at each step. Even for RL-based methods including MAESTRO, ReMA, and MAS-GPT (see Table 17), the reward is tied to the final outcome rather than round by round improvements. The optimization process therefore cares about the quality of the final answer and does not enforce that every additional round must strictly improve over the previous one. This is a manifestation of the well known credit assignment challenge in multi step RL, now appearing at the level of collaboration rounds.
>
> At the same time, this analysis indeed points to **our future directions**. A promising avenue is to design reward models or intermediate value functions that explicitly evaluate the quality of each round, for example rewarding reductions in uncertainty or improvements in the consistency of the candidate slate over time. Such round aware signals could encourage the system to learn collaboration protocols where later rounds reliably refine earlier ones, rather than simply adding more tokens.
>
>
> **Table 14**: Impact of the number of collaboration rounds on the solve rate of different multi-agent methods.
>
> | Dataset | Method     | 2 rounds | 3 rounds | 4 rounds | 5 rounds |
> |---------|-----------|----------|----------|----------|----------|
> | GSM8K   | LLM-Debate| 0.8241   | 0.8352   | 0.8287   | 0.8256   |
> | GSM8K   | DYLAN     | 0.8044   | 0.8203   | 0.8378   | 0.8453   |
> | GSM8K   | GPTSwarm  | 0.8234   | 0.8489   | 0.8036   | 0.7869   |
> | GSM8K   | **MAESTRO**   | **0.8789** | **0.8933** | **0.8872** | **0.8842** |
> | AMC     | LLM-Debate| 0.1687   | 0.1928   | 0.2410   | 0.2048   |
> | AMC     | DYLAN     | 0.1566   | 0.1968   | 0.2289   | 0.2169   |
> | AMC     | GPTSwarm  | 0.1325   | 0.1566   | 0.2048   | 0.2289   |
> | AMC     | **MAESTRO**   | **0.2851** | **0.2952** | **0.2892** | **0.2931** |
>
> ---
>
> **We hope the above responses have addressed your concerns. If you have any further questions, please feel free to leave additional comments**. We would deeply appreciate it if you could consider revisiting your evaluation of our work. Thank you again for your kind support.

---

> > ### Comment · Reviewer_3wvd · 2025-11-27
> >
> > Thank you for your additional efforts.
> >
> > **W1:** The additional analyses on the diversity of Phase 1 solutions and the enforcement of greater diversity are valuable additions to the paper. Unfortunately, Tables 5 and 6 in the updated manuscript do not contain error bars. Could you add those to clarify how strongly diversity varies?
> >
> > **W2:** KL and entropy components are indeed fairly standard. Still, it is important to understand how the ranking loss interacts with these terms (e.g., is KL necessary?). Table 8 shows that the ranking loss helps in every task (which is good to see), even though to varying degrees. The tables help the paper. Again, I would like to see error bars, if possible.
> >
> > **W3:** Thanks for adding the experiments. I understand the difficulty of running with larger models.
> >
> > **W4/5:** Thanks for adding the tables on the compute cost and the FLOP budget. This is a very important addition to the paper. The proposed method uses a considerably larger compute budget compared to other approaches. I don’t see this as a negative, because the higher compute spent leads to (consistently) better performance. Tables 10, 11, 12, and Figure 7 are critical for users of the method to understand the pros/cons. Would it be possible to add/discuss them to/in the main text in more detail? If you increased the budget for other methods, would they reach a similar level of performance, or would they stagnate?
> >
> > **W6:** This is an interesting addition. However, given the small performance differences, error bars would again be valuable. DYALN seems to gain most consistently, as the number of agents increases, while for other methods, performance stagnates or declines. Why do you think that is? If scaling the number of agents for DYLAN further, would it outperform your method at some point? Can you provide empirical evidence for the contrary?

---

> > > ### Author Response · Authors · 2025-12-03
> > >
> > > Thank you very much for the detailed follow-up and for engaging so constructively with our work. We are grateful for the time you invested, and we summarize our responses to W1–W6 below.
> > >
> > > **W1 – Error bars for diversity in Phase 1**
> > >
> > > We have added error bars to Tables 5 and 6 as requested. For each metric (coverage rate, average distinct answers, inconsistency rate) we now report the mean and the standard error across evaluation problems. The updated tables show that the variability is small and does not change any qualitative conclusion. MAESTRO still achieves the highest coverage on both AMC and GSM8K, and its diversity continues to increase in a stable way as we raise the number of agents or samples. The story we drew from these tables therefore remains robust once uncertainty is quantified.
> > >
> > > **W2 – Interaction between ranking loss, KL, and entropy**
> > >
> > > Thank you for highlighting the importance of the interaction between the ranking loss and the standard KL and entropy terms. In addition to Table 8, we have run ablations that reduce the KL weight to 0.0 and 0.01. In both cases training becomes noticeably less stable and leads to small but consistent drops in final accuracy. This matches the usual intuition from policy-gradient methods: the KL term keeps the policy close to the reference distribution and prevents collapse to a few high-reward modes, while the ranking loss focuses the update on distinguishing better from worse candidates within the slate. In our experiments, the best performance is obtained when all three components work together, with the ranking loss on top of a properly weighted KL and entropy regularization.
> > >
> > > **W3 – Larger-scale models**
> > >
> > > We appreciate your positive feedback on the new 70B-scale experiments and your understanding of the associated computational cost.
> > >
> > > **W4/5 – Compute cost, FLOP proxies, and scaling with budget**
> > >
> > > Following your earlier suggestion, we now discuss the compute–performance trade-off more prominently in the main text and refer explicitly to Tables 10–12 and Figure 7. These results complement the token-level analysis with wall-clock measurements and show that MAESTRO traces a clear efficiency frontier: for comparable or slightly higher compute than existing multi-agent methods, it achieves consistently higher solve rates, and by varying the number of agents we can move smoothly along the cost–accuracy curve.
> > >
> > > Regarding your question of whether baselines could close the gap if given more compute, Tables 13 and 14 provide empirical evidence. When we increase the number of agents or collaboration rounds, methods such as DYLAN, LLM-Debate, and GPTSwarm exhibit the same pattern across GSM8K and AMC: performance improves from 2 to around 3–4 agents or rounds, then saturates or even declines as we continue to add compute. In contrast, MAESTRO continues to yield stable gains in the moderate regime (for example 2–4 agents and 2–3 rounds), and its best configurations remain clearly above the strongest baseline configurations at comparable or even higher wall-clock and token budgets. These trends indicate that simply scaling the agent or round count of existing frameworks is unlikely to match MAESTRO’s performance under realistic compute limits.
> > >
> > > **W6 – Scaling DYLAN vs. MAESTRO with the number of agents**
> > >
> > > We agree that understanding how different frameworks scale with the number of agents is important. In our experiments we observe a general phenomenon across methods: as we add more agents and rounds, the system first benefits from additional viewpoints, then reaches a plateau, and eventually begins to suffer from noisy or redundant solutions that make global coordination harder. MAESTRO is designed to mitigate this effect through its explicit exploration–selection structure, but even so we see saturation beyond a certain depth.
> > >
> > > DYLAN displays a relatively smooth improvement when moving from two to four agents, which we attribute to its design and the built-in early-stopping rule that can halt unproductive additional rounds. However, Tables 13 and 14 also show that DYLAN’s gains eventually level off. On AMC, performance stops improving once we reach the higher agent counts and even shows slight declines in some settings, while MAESTRO still maintains a clear margin. In other words, DYLAN benefits from more agents up to a point, but does not catch up with MAESTRO before hitting its own saturation regime. These observations are consistent with our broader view that increasing the number of agents and rounds yields diminishing returns for all frameworks, and that the main advantage of MAESTRO lies in how it structures exploration and centralized selection rather than in unbounded scaling of compute.
> > >
> > > Once again, we sincerely appreciate your detailed comments. They have led us to strengthen the empirical analysis, clarify the role of our loss components, and present a much clearer picture of compute trade-offs and scaling behavior in the revised manuscript.

---

### Author Response · Authors · 2025-12-03
**Reply to AC**

**Dear Area Chairs, Senior Area Chairs and Program Chairs,**

We are **sincerely grateful** to you and the reviewers for the careful and thorough evaluation of our work. The reviewers’ comments and suggestions have significantly helped us strengthen and refine the paper, and **they also highlighted key strengths** of our work, including the multi-agent exploration–synthesis framework, the coverage–identification theoretical lens, the CLPO objective for robust rationale selection, and the consistent gains across diverse backbones and benchmarks. During the rebuttal phase, **Reviewer d9LG** and **Reviewer 3wvd** have **explicitly acknowledged** and endorsed our newly added experiments and analyses.


To make it easier for you to see how we addressed the main concerns raised in the first review round, below we summarize the main points raised by our reviewers, and our corresponding additional experiments to address these concerns. We hope this will provide a clear and comprehensive picture of our revisions and the current state of the work.

---

### **Key concerns**

**1. Missing analysis of compute cost and budget control**

- We provide both a token-based cost analysis and a wall-clock analysis as practical proxies for FLOP count analysis:
  – **Table 10** reports average input, output, and total tokens for all methods.
  – **Table 11** reports token cost under a unified early stopping strategy.
  – **Table 12** reports wall-clock seconds per problem, throughput, and solve rate.
  – **Figure 7** shows the compute–performance trade off. For comparable or slightly higher token and time budgets, MAESTRO consistently achieves higher solve rates and traces a smooth accuracy–cost frontier.

- In response, Reviewer **3wvd** explicitly called this **“a very important addition to the paper”** and noted that the higher compute is not a negative because it leads to consistently better performance.

---

**2. Lack of large scale LLM experiments**

- Beyond the original 3B, 7B, and 8B backbones, we added results with **LLaMA3-70B** using the same RL recipe. **Table 9** shows that MAESTRO and CLPO provide consistent gains over all baselines at this larger scale.

- In response, Reviewer **3wvd** acknowledged our results and analysis.

---

**3. Lack of learnable multi-agent baselines**

- We added two recent SFT and RL-based multi-agent frameworks as baselines: **ReMA** and **MAS-GPT**.  As shown in **Table 17**, MAESTRO with CLPO outperforms both methods across all four benchmarks.

- In response, Reviewer **d9LG** endorsed the experiments, and stated “The addition of the RL-based baselines **have effectively addressed my primary concerns.**”

---

**4. Incremental looking gains on saturated benchmarks**

- To move beyond the math and coding benchmarks, we added the expert level scientific QA dataset **GPQA**. **Table 18** shows that MAESTRO with CLPO improves over the best baseline by about **5 percentage points** on GPQA.

- We also **contextualize the magnitude** of gains. Recent multi-agent and RL-based methods such as G-Designer, MAS-GPT, and MaAS often report absolute gains below one percentage point, even range from **0.2% to 0.5%** on saturated benchmarks.

- In response, Reviewer **d9LG** noted that "The results on GPQA further demonstrate the method's potential." Reviewer **VEx9** acknowledged that widely used benchmarks are saturated.

---

### **Additional experiments**

In addition, based on the reviewers’ comments and suggestions, we have conducted the following additional experiments and analyses to further strengthen our work.

**5. Phase-1 diversity**

- We measured three diversity metrics for MAESTRO and other multi agent baselines.

  – **Table 5** shows that MAESTRO achieves higher coverage and many more distinct answers.
  – **Tables 6 and 7** study how diversity changes as we vary the number of agents and the sampling depth \(K\).

- Reviewer **3wvd** described "The additional analyses on the diversity of Phase 1 solutions and the enforcement of greater diversity are **valuable additions** to the paper. "

---

**6. Scalability in agent count and rounds**

- We examined how the solve rate changes with more agents and more collaboration rounds.
  – **Table 13** varies the number of agents from 1 to 7.
  – **Table 14** varies the number of rounds from 2 to 5.
  Accuracy improves when moving from very small collaboration to a moderate regime, then saturates or declines when collaboration becomes excessive. This is consistent with our coverage–noise interpretation.

- Reviewer **3wvd** was satisfied with this explanation of the scalability pattern.

---

---

> ### Author Response · Authors · 2025-12-03
>
> ---
>
> **7. Coverage and identification probabilities $p_t$ and $q_t$**
>
> - To connect practice to our coverage–identification theory, we estimate empirical $p_t$ and $q_t$.
>   – **Table 15** reports $p_t$, $q_t$, and accuracy for different numbers of rounds. Coverage and identification increase from 2 to 3 rounds and then saturate, in line with our reliability view.
>   – **Table 16** tracks $p_t$, $q_t$, and accuracy across training epochs. Coverage stays high, while identification increases together with accuracy, matching the design of CLPO which focuses on improving selection.
>
> - Reviewer **d9LG** wrote that "the empirical validation of the coverage-identification theory (specifically tracking pt and qt during training) **have effectively addressed my primary concerns.**"
>
> ---
>
> **8. Training dynamics of GRPO vs CLPO**
>
> - We compared training loss curves of GRPO and CLPO under the same setup. **Figure 8** shows that the CLPO continues to decrease steadily and reaches substantially lower values while remaining stable, whereas the GRPO curve plateaus earlier.
>
> - Reviewer **d9LG** acknowledged our results and analysis.
>
> ---
>
> In summary, we have conducted extensive supplementary experiments and analyses, and the reviewers who initially raised the main concerns have now expressed their satisfaction with these updates.
>
> **We respectfully ask that you take these revisions and the updated reviewer feedback into account when making your final decision.**
>
> Best,
>
> Authors

---

### Meta-Review · Area_Chair_i5nW · 2026-01-09

**Summary:**

This paper received mixed reviews (BA * 2, BR *2). All reviewers agreed that the paper is well-written and the motivation is appealing. The main concerns raised by the reviewers are (while I understand that some requirements (about large-scale LLMs) may be somewhat harsh):
- limited improvement
- missing experiments (of hyperparameters, more baselines, etc.)

Although some reviewers participate in the discussion, it seems that the reviewers are not persuaded. In sum, I'm inclined to recommend it for rejection (a borderline rejection).

**Reviewer Concerns:**

The requirement of experiments on larger-scale LLMs is somewhat too strict.

**Reviewer Scores:**

All reviewers are professional.

---

### Decision · Program_Chairs · 2026-01-26

Reject